# Altered extracellular matrix structure and elevated stiffness in a brain organoid model for disease

Maayan Karlinski Zur [1,2,3], Bidisha Bhattacharya [1,3], Inna Solomonov [2], Sivan Ben Dror[4], Alon Savidor [5], Yishai Levin[5], Amir Prior [5], Tamar Sapir[1,3], Talia Harris[6], Tsviya Olender[1], Rita Schmidt [7,8], J. M. Schwarz[9], Irit Sagi [2] ✉, Amnon Buxboim [4,10,11] ✉ & Orly Reiner [1,3] ✉

The viscoelastic properties of tissues influence their morphology and cellular behavior, yet little is known about changes in these properties during brain malformations. Lissencephaly, a severe cortical malformation caused by *LIS1* mutations, results in a smooth cortex. Here, we show that human-derived brain organoids with *LIS1* mutation exhibit increased stiffness compared to controls at multiple developmental stages. This stiffening correlates with abnormal extracellular matrix (ECM) expression and organization, as well as elevated water content, measured by diffusion-weighted MRI. Short-term MMP9 treatment reduces both stiffness and water diffusion levels to control values. Additionally, a computational microstructure mechanical model predicts mechanical changes based on ECM organization. These findings suggest that *LIS1* plays a critical role in ECM regulation during brain development and that its mutation leads to significant viscoelastic alterations.

Tissue mechanics is crucial in shaping tissue growth, function, and disease progression. However, this field is understudied in human brain developmental diseases due to limited access to human brain tissues and suboptimal animal models[1–6]. It has been suggested that the tissue composition, the dynamic cellular processes that occur in the developing brain, and tissue mechanics play a role in shaping brain structure[7–19]. The organization, shape, and amount of structural extracellular matrix (ECM) in the tissue provide tissues with their mechanical forces, thereby affecting cellular behavior during and after development[20,21]. ECM has been proposed to be a crucial element in the structural organization of the developing brain and the

formation of human brain folds[22,23]. Lissencephaly, characterized by the absence of cortical convolutions, provides insights into human brain fold formation. LIS1 mutations, prevalent in lissencephaly patients, impact the scaffold protein LIS1, affecting cytoplasmic dynein, RNA interactions, splicing, and gene transcription[24–26]. Having established that LIS1 influences the physical characteristics of embryonic stem cells in our earlier research[27], we sought to investigate the unknown aspect of how these properties are affected at the tissue level. Studying this disease in mouse models has been useful but limited since their cortex naturally lacks convolutions, and human brain organoids, which usually lack folds, have the potential

[1]Department of Molecular Genetics, Weizmann Institute of Science, Rehovot, Israel. [2]Department of Immunology and Regenerative Biology, Weizmann Institute of Science, Rehovot, Israel. [3]Department of Molecular Neuroscience, Weizmann Institute, Rehovot, Israel. [4]The Institute of Life Sciences, The Hebrew University of Jerusalem, The Edmond J. Safra Campus, Jerusalem, Israel. [5]The De Botton Protein Profiling Institute of the Nancy and Stephen Grand Israel National Center for Personalized Medicine, Weizmann Institute of Science, Rehovot, Israel. [6]Department of Chemical Research Support, Weizmann Institute of Science, Rehovot, Israel. [7]Department of Brain Sciences, Weizmann Institute of Science, Rehovot, Israel. [8]The Azrieli National Institute for Human Brain Imaging and Research, Weizmann Institute of Science, Rehovot, Israel. [9]Physics Department, Syracuse University, Syracuse, NY, USA. [10]School of Computer Science and Engineering, The Hebrew University of Jerusalem, The Edmond J. Safra Campus, Jerusalem, Israel. [11]The Alexander Grass Center for Bioengineering, The Hebrew University of Jerusalem, The Edmond J. Safra Campus, Jerusalem, Israel. ✉e-mail: Irit.Sagi@weizmann.ac.il; amnon.buxboim@mail.huji.ac.il; orly.reiner@weizmann.ac.il

to form them, as we have previously demonstrated[16]. The mutant mice exhibited deficits in neuronal migration and hippocampal pathology[28–36]. Here, we used human pluripotent stem cell-derived organoids to study how biomechanical changes are involved in cortical malformation development. We found that brain organoids mutated for *LIS1* are stiffer than control organoids and unraveled a substantial ECM disorganization phenotype in the disease. Using an interdisciplinary approach, we applied data obtained from rheological tests, MRI, and ECM composition and structure characterization to develop a computational model. This model successfully predicts mechanical changes associated with differential ECM localization and integrity in the developing brain.

## Results

### Mutant *LIS1* brain organoids are stiffer

To investigate whether the cortical structural abnormalities observed in cases of lissencephalic pathologies are linked to mechanical abnormalities, we performed rheological tests on cortical organoids (corticOs). These organoids were generated from two types of cell lines: control human embryonic stem cells (hESCs) and isogenic lines with a *LIS1* heterozygous mutation introduced using CRISPR/Cas9 genome editing techniques[16].

CorticOs were generated using a protocol for self-organizing cortical tissue (Supplementary Fig. 1a). A series of characterizations on different days indicated that the early ectoderm-like organoids were expressing different neural progenitors on days 9 and 18; and by day 60 corticOs contained post-mitotic neurons and astrocytes (Supplementary Fig. 1b–e). To enrich the basal radial glial progenitors population, which is thought to play an important role in the development of the human cortex[37], we added hLIF from day 35. Basal radial glial progenitors were detected in 96-day-old corticOs (Supplementary Fig. 1f, g).

Changes in mechanics can alter the structure, development, and function of cells that make up a tissue, such as the brain[38]. To determine the mechanical effects of *LIS1* mutations, we employed micropipette aspiration (MPA) rheology and performed creep test measurements of brain organoids from *LIS1*[+/−] and control organoids at multiple developmental ages (days 9, 18, 35, and 70). The aspiration dynamics of the organoids into the pipette under a constant negative pressure were recorded and analyzed (Fig. 1a). All organoid measurements, regardless of their developmental age, shared a characteristic response to the applied load: organoids stretched elastically the moment suction was applied, followed by a gradual aspiration into the pipette over five to ten seconds, and approached a steady-state finite deformation (Supplementary Fig. 2a).

The mechanical behavior of organoids was analyzed using the standard linear solid (SLS) model. In its Maxwell representation, it consists of an elastic element (spring $k_1$) that is connected in parallel to a second elastic element (spring $k_2$) positioned in series with a viscous element (dashpot $\mu$) (Fig. 1a'). We calculated the creep compliance function, $J(t)$, to measure time-dependent deformability[39], using parameters like the aspirated fraction length, pipette radius, applied pressure, and a geometrical factor. The organoid mechanics were quantitatively characterized by fitting the SLS creep compliance function. We assessed the instantaneous stiffness $k_0$, steady-state stiffness $k_{st}$, and the viscoelastic transition time scale $\tau$. This model showed high accuracy in representing organoid behavior, as evidenced by the high R-square values in our fits (See Methods for a full description of the model and its calculation).

CorticOs stiffness, as estimated by $k_0$ and $k_{st}$, ranged over hundreds of pascals, indicating that it is as soft as cream cheese (Fig. 1b–e)[40]. Notably, this range aligns with the lower spectrum of brain tissue stiffness, which typically spans from 0.1 to 2 kilopascals[41]. With time, the corticOs stiffen, likely due to continuous ECM deposition and/or fibrillation. We found that *LIS1*[+/−] mutations increased the

stiffness of corticOs as early as 9 days after their aggregation and that they remain stiffer than controls up to the latest tested time point, day 70. However, no significant difference in the viscoelastic transition time scale $\tau$ is observed (Fig. 1b'–e'). These findings indicate that, when subjected to a physiologically relevant load across multicellular length scales, the corticOs exhibit characteristics of a solid-dominant viscoelastic behavior that becomes stiffer with progressing developmental stages. Overall, our data suggest that changes to the biomechanics of *LIS1*[+/−] organoids appear early in development and continue over a considerable period.

### Cortical *LIS1*[+/−] organoids express abnormal ECM and more Lamin A

To delineate the molecular changes that are associated with the stiffening of the *LIS1*[+/−] cortices, we analyzed the proteomic signature of control and *LIS1*[+/−] corticOs. On day 35, we extracted proteins from the *LIS1*[+/−] mutated corticOs, WIBR3 control, and a PX335 control. The PX335 control line was created with the empty Cas9 nickase plasmid used in creating the original LIS1 mutant, electroporated into the parental WIBR3 line to produce a second control CRISPR corticOs. A total of 7842 proteins were identified and quantified, of which 429 were differentially expressed (DE) in the mutation, based on the threshold criteria of Log2Fold change ≥|0.5|, at least 1 peptide per protein, and ANOVA *p* value < 0.05 in both PX335 and WIBR3 control vs. *LIS1*[+/−] comparisons (Supplementary Data 1a). We then conducted a Metascape analysis[42] to examine the pathways differing between the two control lines and the mutated corticOs, and identified proteins associated with the ECM as most affected (Supplementary Data 1b, Fig. 2a). In addition, previous studies indicated that basal radial glial cells might be involved in cortical gyrification (review[43]). Therefore, we have chosen to conduct an additional analysis in 105-day-old corticOs, following the appearance and expansion of that progenitor population (Supplementary Fig. 1f, g). On day 105, there were 526 DE proteins between the *LIS1*[+/−] and control corticOs (Fig. 2b, Supplementary Data 2a, b). The top affected pathway of the DE proteins common identified by www.geneanalytics.com[44] was the superpath "collagen-containing extracellular matrix", which was consistent even when using more stringent sorting (Log2Fold change ≥|1|.

In line with the observed stiffening of the corticOs in the MPA procedure, we also observed increased expression of the LMNA protein in the *LIS1*[+/−] corticOs (Fig. 2b). Lamins are intermediate filament proteins of the nucleus that provide structural stability. Lamin A expression is strongly correlated with tissue stiffness, whereas other members of the nuclear lamina family, Lamin B, and Lamin C, are not affected by the physical properties of the cells[45]. The roles of LMNA are not strictly structural as it also affects chromatin organization, gene regulation, cell differentiation, and signaling pathways, including the Wnt/β-catenin pathway, TGFβ, and Notch[46].

The 30% increase in LMNA protein levels indicated by the proteomics was recapitulated by Western blot analysis, whereas the levels of Lamin B were unchanged (Fig. 2c, d). It was further hypothesized that the rise in Lamin A would reduce the double-strand breaks in the tissue due to an increased nuclear protective shield. This was supported by the Western blot quantification, which showed a significant reduction in the double-strand breaks marker γ-H2AX in the *LIS1*[+/−] 105-day-old corticOs. These changes in chromatin and lamins expression may potentially affect tissue stiffness; however, in this work, we focused on the ECM. Overall, the increased levels of Lamin A, together with the changes observed in ECM-related proteins, suggest that a mutation in LIS1 results in adverse biomechanical abnormalities to the brain organoids.

In addition, RNA-seq analysis of the same 105-day-old corticOs cohort revealed a total of 2061 DE genes between the *LIS1*[+/−] and control samples (Supplementary Fig. 3a, Supplementary Data 3). Pathway analysis of DE genes indicated that here, too, the collagen-associated

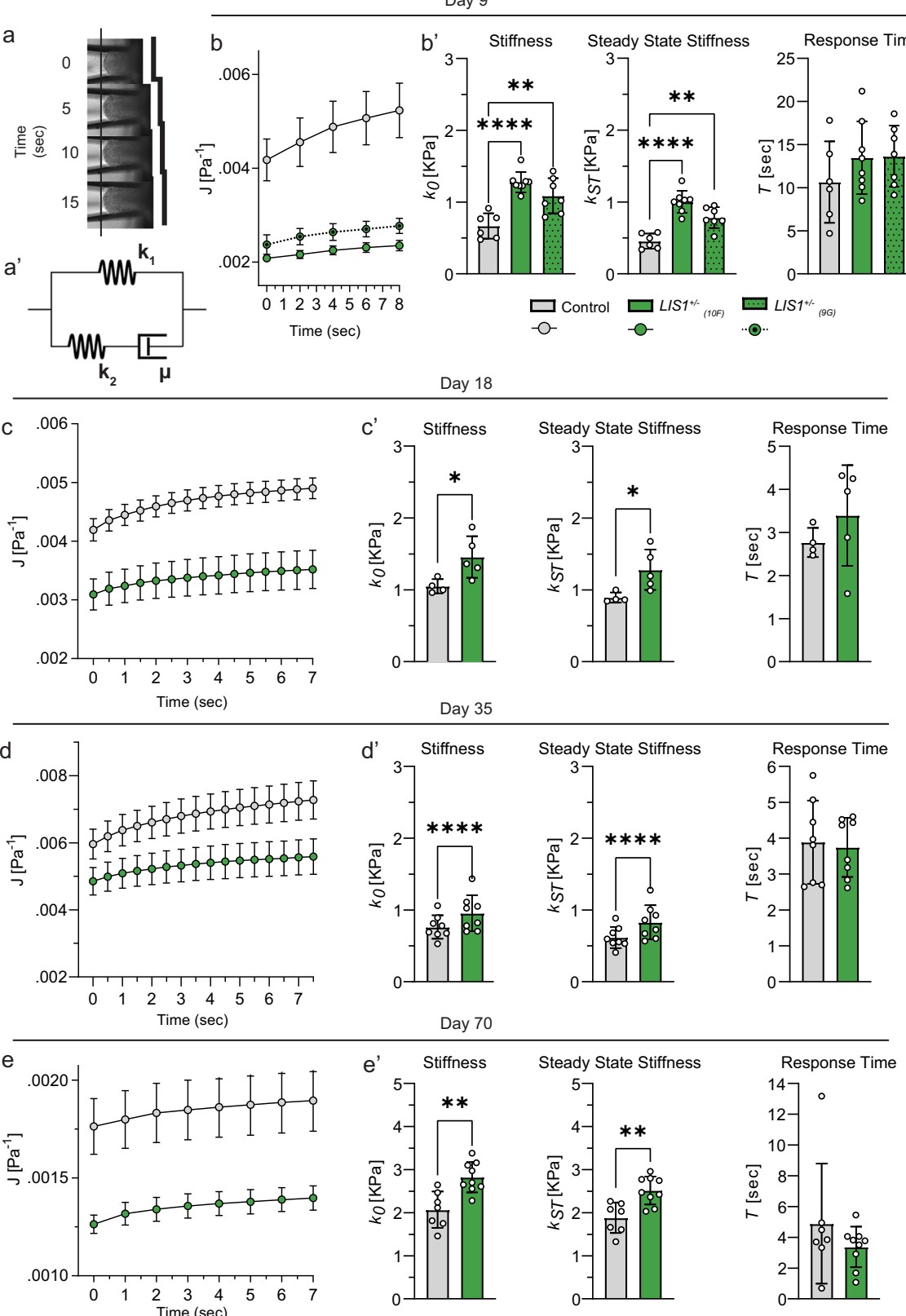

pathway was the most affected GO-term in the mutation (Supplementary Fig. 3b). In addition, we observed no significant increase in the expression of the *LMNA* gene. Accordingly, genes involved in generating, processing, or regulating collagens, such as several non-fibrillar collagens, including *COL12A1*, *COL14A1*, *COL16A1*, and *COL24A1* were affected in *LIS1* mutant corticOs. Other collagen coding genes were downregulated in the mutation, including different α-chains of collagen type IV and *COL5A3*[27,44,47,48].

### Hippocampal *LIS1* organoids exhibit increased ECM

Mutations in *LIS1* substantially impact the organization and structure of the cerebral cortex, and some abnormalities have been noted in

**Fig. 1 | LIS1 mutation leads to the stiffening of brain organoids, which are solid-like viscoelastic tissues. a** Under constant suction pressure, organoids are continuously aspirated into the pipette and gradually approach a steady-state deformation. The organoids' creep compliance is well-fitted by the standard linear solid (SLS) viscoelastic model (**a'**). **b–e** Averaged creep compliance measurements (symbols) are fitted by the SLS model (curves) at the specified conditions. Symbols and error bars correspond to the mean and standard error of the mean. **b'–e'** SLS fits the instantaneous ($k_O$) and steady-state ($k_{st}$) stiffness, and response time ($\tau$) are plotted. Symbols represent individual organoids recorded, while the bar graphs

and error bars correspond to the mean and standard deviation. The analysis included the following number of organoids, each measured separately and fitted independently: **b** Day-9: $n_{control} = 6$, $n_{10F} = 7$, $n_{9G} = 7$. **c** Day-18: $n_{control} = 4$, $n_{10F} = 5$. **d** Day-35: $n_{control} = 8$, $n_{10F} = 8$, and **e** Day-70: $n_{control} = 7$, $n_{10F} = 9$. Statistical significance is evaluated via one-way ANOVA test in **b'** and via two-tailed independent student's $t$ test in **c'–e'**. * To strengthen our findings, we tested an additional $LIS1^{+/-}$ ESCs line annotated as 9G, generated in the same approach discussed in the methods. However, throughout the manuscript, we consistently used the 10F cell line.

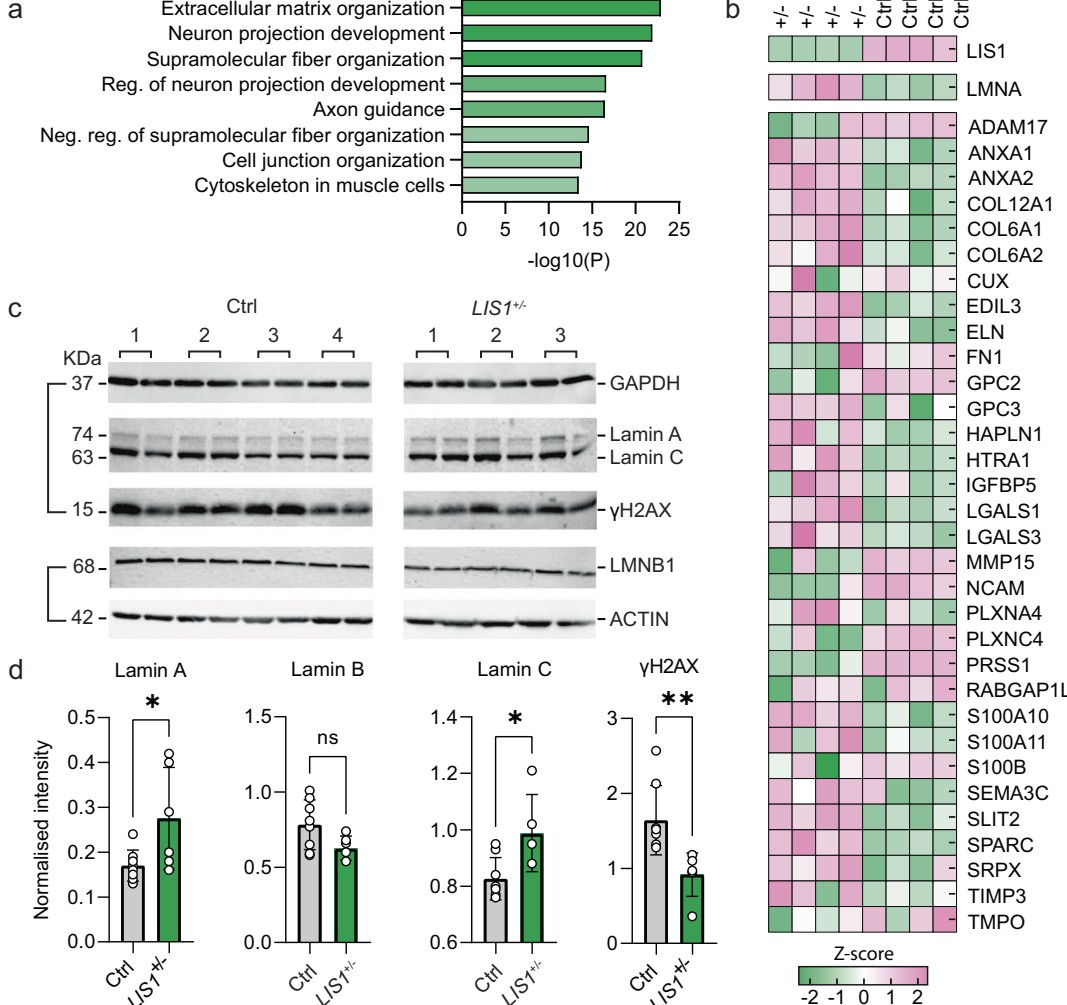

**Fig. 2 | Matrisome composition of corticO. a** Metascape analysis on the proteomics data from 35-day-old corticOs. **b** Heatmap of top DE matrisomal proteins in $LIS1^{+/-}$ and control 105-day-old cortical organoids. Cell color expresses normalized reads following logarithmic transformation and Z-score normalization. **c** Western blot analysis indicated the elevation of Lamin A/C and the reduction in γH2AX in

105-day-old $LIS1^{+/-}$ corticOs. **d** Quantification of western blot results (mean ± SD, two-tailed unpaired student's $t$ test, $\alpha = 0.05$, * means $p$ value < 0.1 and ** means $p$ value < 0.01). The analysis included: **a** Day-35, for each genotype $N_{corticOs} = 4$, $n_{corticOs} = 10–12$, **b, c** Day-105 proteomics and WB: $N_{corticOs} = 4$; $n_{corticOs} = 6–8$.

the hippocampus[49]. The *Lis1* mouse models display pronounced hippocampal abnormalities, but the human hippocampal pathophysiology has not been extensively characterized[28–36]. We generated control and $LIS1^{+/-}$ mutated hippocampal organoids (hippOs). The hippOs were generated by exposing the 18-day-old aggregates to a temporal BMP4 and WNT activation (Supplementary Fig. 4a)[50]. After 70 days, the tissue contained hippocampal-like ZBTB20+ and LEF1+ cells, SOX2+, PAX6+ and HOPX+ progenitors, NeuN+ and MAP2+ neurons, and GFAP+ astrocytes (Supplementary Fig. 4b–h).

Using Mass spectrometry, we explored differences in the protein content of control and mutant hippOs[16] (Supplementary Fig. 4i). A total of 5068 proteins were identified and quantified (Supplementary

Data 4), of which 1178 were DE in the mutation, based on the threshold criteria of $p < 0.05$, Log2Fold change ≥|0.5|, and >1 peptide per protein. The proteome confirmed a considerable reduction of the mutant LIS1 protein and revealed that $LIS1^{+/-}$ organoids were highly enriched with ECM-related proteins (Supplementary Fig. 4i). DE ECM proteins included an increased level of structural proteins in the mutant, including several collagen types. These excessive collagens undergo efficient hydroxylation in the mutant organoids, as assessed by the post-translational modifications (PTM) analysis extracted from the mass spectrometry data (Supplementary Fig. 4j, Supplementary Data 5). Increased levels of the enzymes involved in these modifications (P3H1-3 and PLOD-3) in the mutant samples

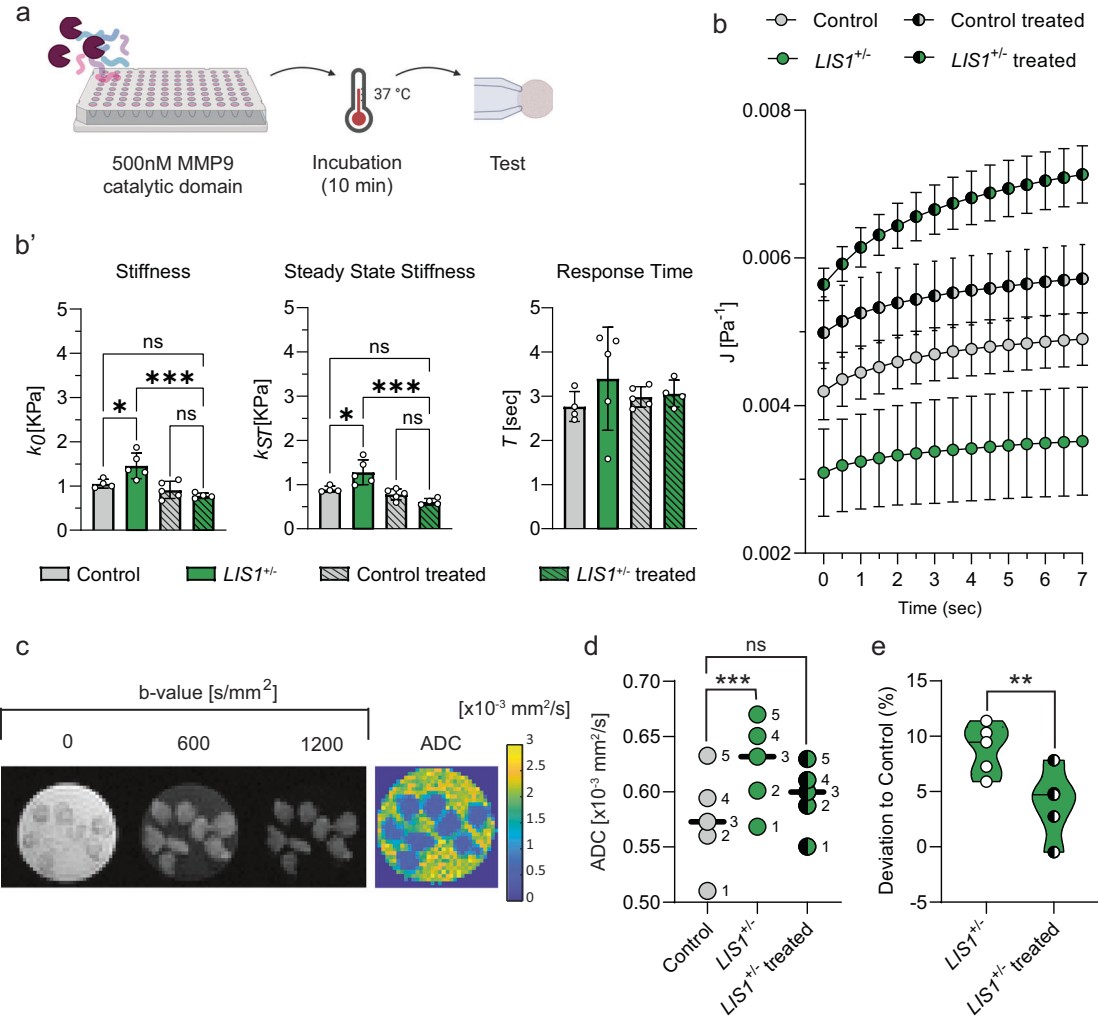

**Fig. 3 | The effects of *LIS1*⁺/⁻ mutation and ECM proteolysis on corticOs mechanics and structural organization. a** Day-18 control and *LIS1*⁺/⁻ mutated organoids were treated with 500 nM MMP9 catalytic domain for 10 min at 37 °C before being submitted for MPA creep test measurement and MRI imaging. Created in BioRender. Solomonov, I. (2025) https://BioRender.com/bncn17s. **b** Similar to the non-treated organoids, the creep compliance of MMP9-treated organoids of both genotypes was fitted to the linear viscoelastic SLS model with a high goodness of fit (R-square ($R^2 > 0.99$)). Symbols and error bars correspond to mean and SEM across, organoids. **b'**, Mean ± SD of the fitted SLS viscoelastic elements, $k_O$, $k_{st}$, and $\tau$ show greater softening of the mutated corticOs by ECM proteolysis. Statistical significance is evaluated via one-way ANOVA test with the following number of organoids, each measured separately and fitted independently: $n_{control} = 4$, $n_{10F} = 5$, $n_{controlMMP9} = 5$, $n_{10FMMP9} = 4$. **c** A representative DW-MRI of eight *LIS1*⁺/⁻ corticOs (left) and the calculated ADC map. One central slice is shown. Three out of six b-values (the degree of diffusion weighting) are shown for a central slice. **d** Estimated maximal likelihood position of the ADC (apparent diffusion coefficient) distributions of control, *LIS1*⁺/⁻, and MMP9-treated *LIS1*⁺/⁻ corticOs show the effect of altered genotypes rescued by ECM proteolysis. Error bars correspond to SD across consecutively repeated scans. **e** The relative deviations of the ADC values of *LIS1*⁺/⁻ and MMP9-treated *LIS1*⁺/⁻ corticOs from control organoids ($n = 5$) are plotted. The potential impact of longitudinal drift was eliminated by calculating each scan's relative deviation, which was then averaged.

further support this. Co- and post-translational hydroxylation of proline residues are essential for maintaining the stability of the triple helical collagen structure, which plays a crucial role in brain development by influencing the properties of the ECM, as well as interactions between cells and their nearby collagens, thus impacting cell behavior[51–53].

In addition to the enrichment in structural proteins, we also observed increased levels of ECM remodeling proteins, such as MMP-14 and LOX, as well as serine protease inhibitors such as SERPINA3, SERPINE2, SERPINB6, and others. These observations were also supported by Gene Set Enrichment Analysis[54], which highlighted the enrichment of genes associated with ECM, collagen fibril organization, and more (Supplementary Fig. 4k). Overall, these findings suggest that brain organoid models representing the human hippocampus also demonstrated pathology linked to the

mutation. This aligns with the limited human data available and is in line with observations from previous mouse models.

## Rescue treatment with MMP9 softened the *LIS1*-mutated organoids

The stiffening effect of *LIS1*⁺/⁻ mutations can be explained by the altered regulation of ECM secretion and remodeling, as demonstrated by our mass-spec data. To examine whether the stiffness is derived from excessive structural ECM proteins, we treated day 18 corticOs with the catalytic subunit of MMP9. This zinc-dependent endopeptidase can cleave multiple ECM and non-ECM fibers broadly expressed in the brain[55,56]. The activity of the MMP9 catalytic domain was tested first at different concentrations by an ELISA activity assay (Supplementary Fig. 5a). CorticOs were then immersed with 500 nM MMP9 catalytic domain for 10 min and submitted for MPA rheology (Fig. 3a). Despite

the proteolytic treatment, the organoids maintained an SLS-like aspiration dynamics and the creep compliance functions indicated a significant increase in deformability (Fig. 3b). Indeed, MMP proteolytic treatment decreased both $k_0$ and $k_{st}$ by -15% in non-mutated organoids and -50% in $LIS1^{+/-}$ mutated organoids relative to non-treated organoids. Notably, the response time $\tau$ remained invariant to ECM digestion (Fig. 3b').

In addition, it has been previously shown that mutations cause biomechanical changes at the cellular level[16]. So far, we found that $LIS1^{+/-}$ organoids were stiffer ($k_0$ and $k_{st}$) compared to control organoids and that this stiffness was reduced by MMP9 treatment. However, MPA tests with 12.5 μm pipettes, in which cell-level mechanics can be examined, revealed that cell contractility interference by myosin ATPase inhibition of blebbistatin also resulted in softened $k_o$ and $k_{st}$ parameters in 18-day-old $LIS1^{+/-}$ organoids (Supplementary Fig. 5b–d). $LIS1^{+/-}$ organoids exhibit prolonged energy dissipation, likely due to increased ECM deposition. These findings confirm that the $LIS1$ mutation enhances stiffness and friction at both the cell and tissue levels. Combined, the MMP9 and blebbistatin treatments implicated both ECM and contractile forces involved in tissue stiffening of the $LIS1^{+/-}$ organoids.

The effects of $LIS1^{+/-}$ mutations and ECM proteolysis on organoid mechanics are likely associated with structural remodeling. To test this directly, we employed diffusion-weighted magnetic resonance imaging (DW-MRI) to compare differences in the structural organization of day 18 control, $LIS1^{+/-}$, and MMP-treated $LIS1^{+/-}$ corticOs. Notably, DW-MRI was previously shown to provide noninvasive means for identifying changes in ECM organization[57], extracellular free water component[58], stiffness, and other structural parameters[59]. To examine the sensitivity of DW-MRI, groups of control, $LIS1^{+/-}$, and MMP-treated $LIS1^{+/-}$ containing multiple corticOs were placed in separate wells and scanned (Fig. 3c-left). Taking advantage of the ultra-high magnetic field (15 T MRI scanner), we reached a high imaging resolution of $100 \times 100$ μm² in-plane and 200 μm slice thickness voxels. Apparent diffusion coefficient (ADC) maps were calculated (Fig. 3c-right), and the maximal likelihood position was estimated for each group (Fig. 3d). To improve statistical significance, five consecutive scans were performed and analyzed by a matched $t$-test; the matching was effective ($P < 0.0001$). The ADC of $LIS1^{+/-}$ organoids exhibited a statistically significant 8.9 ± 2.2% increase in comparison to control organoids (adjusted $p$ value: 0.0007, $n = 5$) (Fig. 3e). In contrast, the ADC of MMP-treated $LIS1^{+/-}$ organoids was not significantly different from controls. To support our findings, we performed three additional experiments that showed similar trends (Supplementary Fig. 6a–d). These results demonstrate that $LIS1^{+/-}$ mutations contribute to increased organoid stiffness through altered ECM regulation, as evidenced by mass-spec data and the significant impact of MMP9 proteolysis on $LIS1^{+/-}$ corticOs. DW-MRI further validates these findings by highlighting structural changes in ECM organization and stiffness in mutated and treated organoids.

## MMP9 treatment reversed numerous abnormal gene expression patterns in $LIS1^{+/-}$ organoids

We enquired whether some of the DE genes in $LIS1^{+/-}$ would reverse in response to the treatment and its impact on the levels of stiffness and diffusion in the mutated organoids shortly after the treatment. To emphasize potential rescue mechanisms, our attention was directed toward genes that exhibited differential expression between $LIS1^{+/-}$ and control groups prior to treatment but not afterward (Supplementary Data 6). A heatmap of a subset of these genes is shown in Fig. 4a. These results indicate that at least part of the response is mediated through immediate (-10 min) changes in gene expression. One possible mechanism for this quick recovery is a stiffness-sensitive expression of micro-RNAs (miRs).

Previous studies have shown that changes in the stiffness of a substrate used to grow cells affected the expression of a large group of

miRs, and changes in miRs can affect the physical properties of cells and tissues[60–62]. In addition, our group recently reported that $LIS1$ is involved in regulating gene expression at several levels, including gene transcription, RNA splicing, and regulation of miRs. These changes were either dependent or independent of the Argonaute complex[27]. Thus, we also conducted a small RNA-sequencing analysis, which revealed 274 DE miRs (Supplementary Fig. 7, Supplementary Data 7). These results suggest that at least some of the ECM expression abnormalities are linked to altered regulation of these genes' expression by miRNA.

Using our existing database of the miRs and mRNA expression of corticOs on day 105, we found that, indeed, the most significant interaction between DE mRNAs and miRs in the $LIS1^{+/-}$ mutation was related to abnormal expression of genes associated with the ECM and the miRs that are predicted to target and regulate their expression. Interestingly, when DE miRs identified in small RNA-seq were paired with DE mRNAs to identify regulatory mechanisms supported by both expression profiles, KEGG pathways enrichment analysis (http://microrna.gr/miRPathv3)[44,47,48] pointed to the "ECM-receptor signaling" pathway ($p < 0.0001$) (Fig. 4b, Supplementary Data 8). Targeted genes highlighted by this miR-mRNA comparison included *COL4A3/4/5, COL5A3, ITGA1, COL1A1, THB2, COL24A1*, and *COL6A6* (Fig. 4c). These results suggest that at least some of the ECM expression abnormalities are linked to altered regulation of these genes' expression by miRs.

## Collagen organization and a microstructure mechanical model

To further characterize the ECM abnormality, control and mutated organoids were immunostained for several of the abnormally expressed structural ECM components in mutated and control organoids. While these immunostainings did not record the collagen enrichment, they revealed substantial disorganization of type 4 and type 3 collagen fibers in 9- and 18-day-old $LIS1^{+/-}$ organoids. Whereas an apparent ring-like structure was observed in the control samples, the signal in the $LIS1^{+/-}$ samples was diffuse and punctate (Fig. 5a, b).

These changes were further recorded in Sholl analysis, measuring the frequency of signals at each normalized distance from the center of each organoid on days 9 and 18. On day 9, collagen type 4 forms a ring-like structure near the outer borders of the organoids in control samples but rather scattered in the mutant ($p$ value = 0.0057) (Fig. 5c). By day 18, the ring becomes more internalized in the control organoids but remains more widely and randomly spread in the mutated organoids ($p$ value = 3.28e$^{-47}$) (Fig. 5d). Thus, across different developmental time points, the $LIS1^{+/-}$ organoids continue to exhibit a disorganized distribution of COL4A1 rather than a circularly organized collagen as observed in the controls. These findings suggest that the LIS1 protein is essential for the proper organization and localization of the ECM. We compared the $LIS1$ mutant organoids with the control ones using additional immunostainings, and they had a greater number of SOX2+ and PAX6+ progenitors (Supplementary Fig. 8a, b). However, they did not differ in the population of pVim+ radial glia or HOPX+ basal radial glia (Supplementary Fig. 8c, d). We did observe an increase in the number of pHH3+ mitotically active cells and the number of cells exiting the cell cycle (KI67−/EdU+), which may be attributed to the roles of LIS1 in cell cycle progression; but no changes were noted in the population of KI67+ proliferating cells and EdU+ cells entering the S-phase. Furthermore, there were no significant differences in the number of apoptotic cells (Supplementary Fig. 9).

Finally, given the changes in the amount and spatial organization of the ECM between the control case and the $LIS1^{+/-}$ case, as well as the differences in stiffness between the two cases, we devised a more detailed mechanical model beyond the SLS model to help provide a quantitative link between the amount and organization of the ECM to brain stiffness. More specifically, we considered a computational model that includes both ECM fibers and cells and computed the stiffness of such a model. For simplicity, we investigated a two-

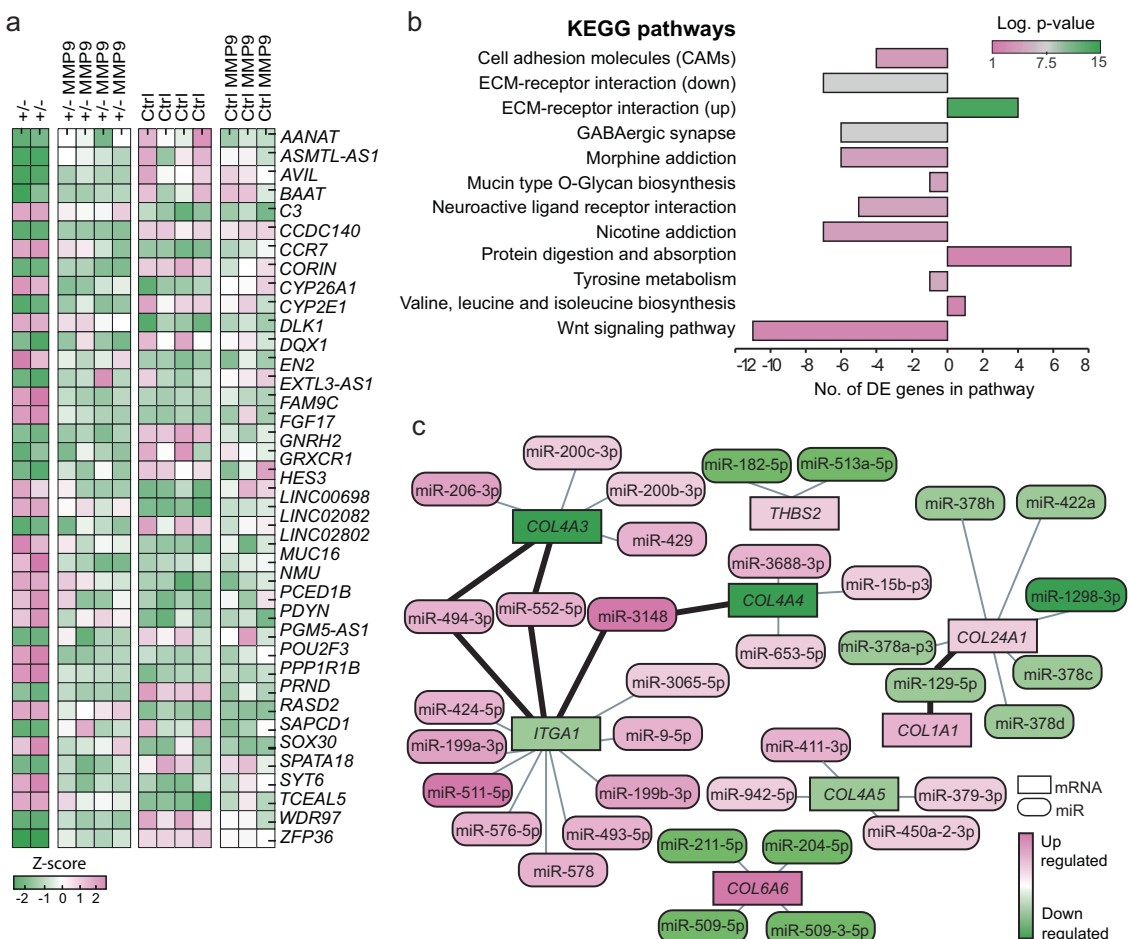

**Fig. 4 | Rescued genes and inverse interaction between mRNA and miRs in _LIS1_<sup>+/−</sup> organoids. a** A heatmap showing the top rescued genes and their expression before and after the MMP9 treatment, $N = 4^*$, $n = 12–14$. **b** Top KEGG pathways identified to be most influenced by the inverse relation between miRs and mRNA expression. Negative numbers in the bar plot indicate that mRNA levels were reduced and that their targeting miRs increased. In contrast, positive values on the x-axis indicate an increase in mRNAs in the _LIS1_<sup>+/−</sup> samples while their targeting miRs were reduced compared to the control. **c** Small RNA-seq integrated with target matrisome-related genes showing the opposite expression pattern of miRs and their predicted targeted mRNAs. *Two outlier groups from the non-treated _LIS1_<sup>+/−</sup> samples were excluded from the analysis.

dimensional model, or a cross-section, of the brain organoid. We will address this simplification in terms of what results we anticipate will carry over to three dimensions below.

The ECM was modeled as a triangular network of fibers, each consisting of both stretching energy and bending energy, with the latter encoding the semiflexibility of the collagen fibers[63,64]. In contrast, the cells are modeled as individual triangles randomly inserted in the lattice with some area stiffness[65] (Fig. 5e–h). While the shapes of the cells in the computational model do not reflect the actual cellular shape, they do encode a key role in the mechanics of the cells. A line of edges on the lattice in one of the three principal directions represents a collagen fiber. To illustrate the disordered structure of the ECM, each edge in the triangular lattice was occupied independently and at random with occupation probability $p$, which is also a measure of the fraction of edges in the network. In other words, the larger the $p$, the more ECM and vice versa, and so changing this $p$ parameter encodes the amount of ECM.

Moreover, given the observation of more patterned ECM in the control case, we also explored an ECM of a similar amount of ECM (similar $p$). However, the ECM is removed only within a localized region (Fig. 5e, f). We then compared patterned ECM with randomized ECM in terms of influencing brain organoid stiffness. We note that such a model has already qualitatively captured nontrivial mechanical features of a fiber network with embedded particles, such as the

phenomenon of compression stiffening[65,66] (See Methods for a detailed description of the model and accompanying numerical calculations).

In control samples (Fig. 5e, f), as the brain organoid tissue is compressed with an increasing amount of uniaxial strain, the cells and fibers begin to distort; therefore, the energy of the tissue increases. We observed that the stiffness of the tissue, measured in simulation units, decreased as more ECM was removed from the localized circular region (Fig. 5i), as one goes from the gray box to the gray X. Notably, the decrease in stiffness is nonlinear, and the ECM provides the tissue with structural support and enhanced stiffness.

However, in the mutant case (Fig. 5g, h), where the ECM does not appear as organized and so the ECM randomly occupies edges on the network, we found that the tissue stiffness decreased nonlinearly as more ECM is removed (See Fig. 5i) as one goes from the green box to the green X. Interestingly, when the patterned ECM stiffness is compared to the random ECM stiffness for the same amount of ECM, the patterned ECM stiffness is not as large as the random ECM stiffness, at least for smaller amounts of ECM, i.e., smaller $p$. We argue that to compare the experiments to our computational model qualitatively, one can go from a smaller amount of ECM (lower $p$) on the patterned curve to a higher amount of ECM (higher $p$) on the random curve as indicated on (Fig. 5i) by going from the gray box to the green box. For instance, if we begin at

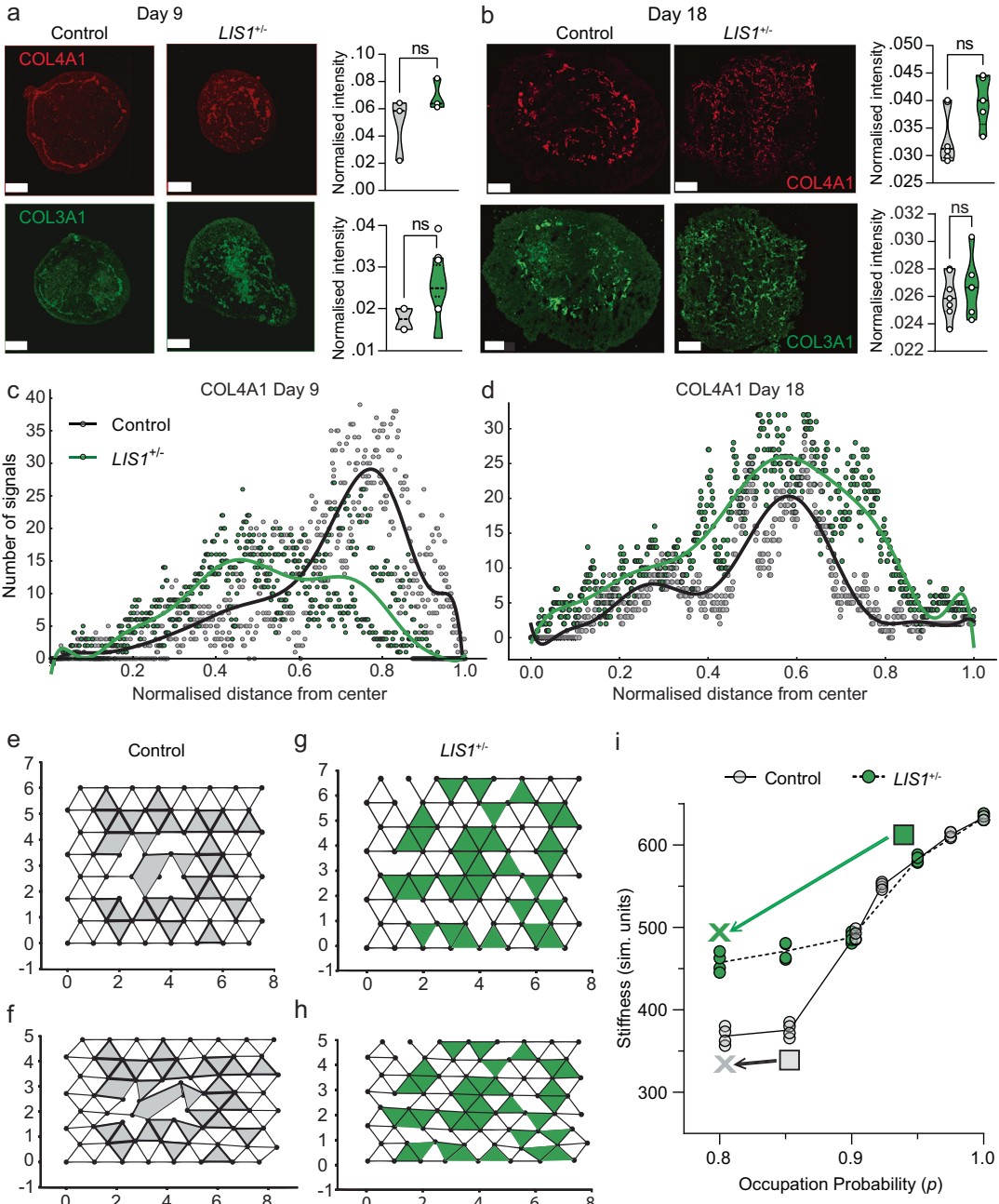

**Fig. 5 | Collagen organization and a microstructure mechanical model.**
**a**–**b** Immunostainings of COL4A1 and COL3A1 on **a** day 9 and **b** day 18, in control and *LIS1*⁺ᐟ⁻ organoids with respective normalized intensity quantification of staining showing no difference between control and mutant organoids (Two-tailed independent student's *t* test, α = 0.05, Day 9: $N_{control}$ = 3, $N_{LIS1+/-}$ = 3; Day 18–$N_{control}$ = 5, $N_{LIS1+/-}$ = 5). Scale bars represent 50 μm. **c**–**d** Sholl analysis of COL4A1 signal in **c** day 9 (Mann–Whitney *U* test for the signal data between control and *LIS1* groups, *p* value = 0.0057) and **d** day 18 (Mann–Whitney *U* test for the signal data between control and *LIS1* groups, *p* value = 3.28e⁻⁴⁷) control and *LIS1*⁺ᐟ⁻ organoids represented as distribution from the center of the organoids highlighting the circular arrangement of collagen deposition in the control compared to an abnormal collagen distribution in the *LIS1*⁺ᐟ⁻ organoids (Mann–Whitney *U* test, α = 0.05). The Sholl analysis included: Day 9–$N_{control}$ = 2, $N_{LIS1+/-}$ = 3; Day 18–$N_{control}$ = 3, $N_{LIS1+/-}$ = 3. **e**–**f** Simulation snapshots for 1% and 20% uniaxial compression strain in the control case. The control case is described by ECM removed within a localized circular region in the center of the system, as motivated by the ring of ECM. **g, h** Simulation snapshots for 1% and 20% uniaxial compression strain in *LIS1*⁺ᐟ⁻ mutant case. The mutant case is described by a randomly diluted ECM. **i** Organoid stiffness (in simulation units) as a function of occupation probability *p*, which indicates the amount of ECM. Here, $N$ = 64, $N_c$ = 32, and $R$ = 10.

*p* = 0.85 fraction of edges occupied on the patterned ECM curve to 0.95 fraction of edges on the random ECM curve, we find a relative increase in the average stiffness of ~54%.

Moreover, this microstructure mechanical model qualitatively supports the results from the MMP9 treatment, in which there was a greater decrease in organoid stiffness for the mutant case compared to the control. As decreasing the occupation probability also reduces the length of ECM fibers, the same trend is observed in our microstructure mechanical model, as indicated by going from the green box to the green X versus going from the gray box to the gray X. Note that this treatment observation is more specific to the parameters at hand. These results were further tested under two additional cases of the parameter space, with more cells and with more stretchable fibers (Supplementary Fig. 10a, b, respectively).

Overall, the computational model supports our findings that an upregulation of ECM enhances stiffness nonlinearly, which is not unexpected. However, our finding that the patterning of the same amount of ECM−diluted from a localized region as compared to randomly−affects organoid tissue stiffness is rather nontrivial in living tissue. The patterning creates weak spots, which, in turn, create a weaker living material. Clearly, the computational model provides a quantitative understanding of the trends observed in the experiments, including the MMP9 treatment findings. Finally, the computational model can predict brain organoid stiffness for decellularized material, presumably leading to compression softening[65,67]. The model can also provide predictions for the trend in any stiffness change for other mutated brain organoids should the organization of the ECM be altered in a different way from the mutant studied here.

## Discussion

Here, we studied the effects of LIS1 haploinsufficiency on ECM composition and organization and how these changes translate into modified physical properties of human brain-like organoids at different developmental time points. Our findings highlight LIS1 as an important regulator of ECM dynamics during human brain development. In both hippocampal and cortical organoid models, *LIS1*[+/−] mutations have been shown to significantly influence ECM composition and organization, leading to observable changes in tissue stiffness. Mass spectrometry analyses confirmed the enrichment of ECM-related proteins, particularly collagens, in *LIS1*[+/−] organoids, indicating a mutation-driven alteration in ECM secretion and remodeling. These proteomic changes were complemented by biomechanical assessments using MPA rheology, revealing increased stiffness in *LIS1*[+/−] organoids. Intriguingly, the application of MMP9, a zinc-dependent endopeptidase, on cortical organoids demonstrated a notable reduction in stiffness, especially in *LIS1*[+/−] organoids, suggesting the reversibility of these changes through ECM proteolysis. Furthermore, DW-MRI provided noninvasive insights into the structural organization alterations in these organoids, reinforcing the link between *LIS1* mutations, ECM dysregulation, and brain tissue mechanics. These results collectively highlight the critical impact of LIS1 on ECM regulation and its subsequent influence on brain development and structural integrity.

The ECM is not merely a passive substance between cells but an active source of signaling and regulating the stem cell niche[68–70]. It consists of molecules secreted by cells, serving as a structural scaffold while also harboring water reserves, growth factors, morphogens, and various bioactive molecules that engage neighboring cells and influence signaling processes[71,72]. Past studies identified that the ECM plays a pivotal role in the development of the nervous system. For example, the synchronized movement of the neural plate and the mesoderm relies on the interplay between two ECM components, laminin and fibronectin[73]. Furthermore, the ECM is highly abundant within neuronal progenitors, and evidence shows that ECM enrichment is most pronounced in progenitors, which is uniquely prevalent in the developing human brain compared to the mouse brain[22,23]. Ex vivo experimental changes in the ECM affected the structure of human embryonic brain sections[20,21]. Our analyses revealed an abundance of matrisome-related proteins in *LIS1*[+/−] hippocampal and cortical organoids, whereas the composition of these two types of organoids exhibited distinctions. Previous studies have indicated that brain regions have ECM content variability[74]. The observed proteomic changes in the corticOs partially correlated with corresponding transcriptomics data (mRNA and small RNA), underscoring the roles of LIS1 in post-transcriptional regulation[16,24,27]. LIS1 is an RNA-binding protein involved in post-transcriptional regulation and governs the physical properties of embryonic stem cells[27].

The *LIS1* mutation affected the physical properties of the brain-like organoids at multiple developmental stages; the *LIS1* mutant organoids were stiffer. The nuclei embedded within this more rigid tissue exhibited modulated parameters. *LIS1*[+/−] brain organoids exhibited increased levels of the nuclear lamina proteins, Lamin A/C, which are known to scale with increased stiffness[39,45,75–77]. In a correlative manner, the levels of DNA damage, indicated by γH2AX, decreased. The decrease in γH2AX levels in LIS1 mutants is significant, as γH2AX marks DNA damage and recruits repair machinery to maintain genomic stability. Reduced γH2AX suggests impaired DNA damage response, potentially hindering effective repair and leading to mutation accumulation. This may exacerbate developmental defects in highly dynamic tissues like the developing brain. The elevated levels of the ECM and increased rigidity are echoed in the ADC values. ADC values reflect the free water content of the tissue, which, in the case of brain-like tissue, is correlated with ECM[78]. ADC values decrease in pathologies that involve cell swelling (edema) and can increase in chronic phases of stroke or other diseases involving necrotic cell death[78,79]. A monkey model employed to simulate a developmental brain disorder known as maternal immune activation, considered a model for autism spectrum disorders, displayed a significant elevation in the presence of extracellular free water within the gray matter of the cingulate cortex[58].

The physical measurements encompassing rheology and MRI appear to have greater sensitivity than the protein-based assays involving mass spectrometry and specific collagen immunostainings. This heightened sensitivity is evident in the fact that we failed to observe a substantial rise in matrisome proteins through proteomics analysis or, in particular, collagens through immunostaining for the early time points. However, immunostaining unveiled an unanticipated spatial arrangement of COL4A1 and COL3A1. While in control samples, they formed a circular structure, in *LIS1* mutant samples, they appeared scattered.

To consolidate our findings, we employed a computational model that takes into account the mechanics of cells, the ECM, and ECM organization. The control and the *LIS1* mutant differed in the amount of ECM and ECM organization and, hence, altered brain organoid stiffness, with the control case being less stiff than the mutant case in both the experiments and the computational model. Moreover, both the experiments and the computational model showed that effectively chopping up ECM enzymatically via MMP9 treatment led to a decrease in stiffness, with a pronounced decrease in the mutant case. Indeed, the mechanics and structure of brain organoids are intertwined as one helps determine the other. Computational models such as the one presented here, as well as other mechanical models[80–84] are, therefore, key to understanding the structure of brain organoids in both healthy and diseased states.

Moreover, multiscale, computational modeling tying the chromatin scale to the brain organoid or tissue scale to unravel the multiscale mechanics-structure relationship is on the horizon[85]. Indeed, examining brain organoids from a material point of view, as demonstrated here, provides perspectives about their structure and will help unravel the intricate mechanics-structure-function relationship in the developing brain and morphogenesis more generally.

Our study reveals the critical role of tissue mechanics in brain development, demonstrating how viscoelastic properties are altered in lissencephaly caused by LIS1 mutations. Using human brain organoids, we uncovered key mechanical and molecular changes, such as increased stiffness and abnormal ECM organization, providing insights into previously understudied aspects of the disease. Combining rheological measurements, MRI imaging, molecular analyses, and computational modeling, we showed that short-term MMP9 treatment can reverse stiffness and water diffusion abnormalities, highlighting the therapeutic potential of targeting ECM dysregulation. These findings establish tissue mechanics as a vital factor in brain malformations and a promising target for intervention. Our research sets a paradigm

for studying mechanical principles in brain disorders, advancing mechanobiology in neurodevelopmental and neurological diseases, while offering innovative therapeutic targets.

## Methods

### Cell lines

An NIH-approved hESC line NIHhESC-10-0079, WIBR3 (W3), was used in this study. Isogenic mutant cell-line clones were previously generated by CRISPR-Cas9-mediated heterozygous deletion in the *LIS1* gene[16]. Cell lines were regularly checked for mycoplasma contamination. The pX335 plasmid, an empty Cas9 nickase plasmid used in creating the original *LIS1*[+/−] cell line, was electroporated into the parental WIBR3 line to produce a second control. The PX335 line was used for the proteomic analysis on 35-day-old organoids. Finally, a single colony was isolated from the WIBR3 line through subcloning and was used as an additional control for the rheology experiment on day 35.

### Generation of hESC-derived corticOs and hippOs

hESCs were cultured in naïve media[86] until 70% confluency and then dissociated and aggregated in low-adhesion wells (Day = 0). Aggregates were primed towards a neurogenic fate by application of the TGF-β- and WNT- signaling inhibitors, SB-431542 and IWR-1, respectively (as specified[50,87]). These molecules facilitate neuroectoderm fate by inhibiting the mesodermal-promoting Nodal/Activin pathway[88]; while preventing premature neuronal differentiation by inhibiting the WNT pathway via AXIN2 stabilization[89,90]. On the 19th day, aggregated neural precursors were introduced to conditions promoting either hippocampal or neocortical formation. In corticogenesis, the region of the dorsal pallium gives rise to the neocortex. In contrast, the medial pallium generates the hippocampus, positioned between the neocortex and the midline cortical hem. The cortical hem secretes dorsalising patterning morphogens such as WNTs and BMPs, which promote medial pallium formation. These also suppress the expression of FGF8 from the competent cortical primordium, which defines a more dorsal neocortical identity[91]. Timely exposure (72 h) to BMP4 and the WNT agonist CHIR-99021 was shown to be sufficient for inducing hippocampal fate by inhibiting GSK3β and activating the WNT pathway[89,90], and was therefore used to generate hippocampal organoids (hippOs). For developing cortical organoids (corticOs), aggregates were grown in a serum-free floating culture where they preferentially expressed FGF8. FGF8 then promotes self-organizing rostral-caudal polarity. After 35 days, corticOs were treated with hLIF to induce bRG progenitor proliferation[37]. Organoids were not embedded or exposed to Matrigel.

### Immunohistochemistry

Ectoderm-like organoids were examined using immunohistochemistry 9 and 18 days after aggregation. Organoids were washed in PBS for 10 min at RT thrice, fixed for 30 min in 4% PFA, and washed again in PBS. Samples were dehydrated overnight at 4 °C in 30% sucrose in PBS, embedded in OCT blocks, and sliced into 16 μm cryosections. Antigen retrieval was conducted in a water bath heated to 90 °C in 10 mM sodium citrate buffer pH = 6 for 20 min and then chilled at RT. Tissues were permeabilized and blocked in a blocking solution (10% normal donkey serum in PBST [0.1% Triton X-100]) for 3 h and then incubated with primary antibodies for 48 h at 4 °C. After three washes, slides were presented with the secondary antibodies from the relevant species to fit the primary antibody for 1:30 h at RT at 1:200 concentration, after which they were incubated with 1:5000 4′,5-diamidino-2-phenylondole (DAPI) for 5 min. All the antibodies used in the study, their dilutions, and catalog numbers are listed in the supplementary information (Supplementary Table 1).

### Image analysis and quantification

Immunostained slides were imaged with Andor Dragonfly spinning disk confocal microscope system HC FLUOTAR ×25/0.95 W VISIR lens,

and stitched to get complete images of the whole organoid sections using the Andor: Oxford Instruments Fusion Shell software. The images were then analyzed using the Spots Analysis of the Oxford Imaris software to determine the percentage of the total number of cells that expressed a particular cell-type-specific marker and compare this parameter between control and mutant organoids. The total number of cells is considered the same as the total number of DAPI[+] nuclei. Markers with nuclear expression, such as KI67, pHH3, EdU, SOX2, PAX6, HOPX, and pVim, were determined by setting up a colocalization filter under the Spots analysis.

For markers like COL3A1 and COL4A1, the mean intensity of the signal in the total area of the organoid section was calculated in Fiji ImageJ software to compare their levels of expression. These values were then compared using statistical tests in GraphPad Prism to determine statistically significant differences, if any.

**Sholl analysis quantification.** The distribution of COL4A1 signal in day 9 and day 18 corticOs was determined using the Sholl analysis plugin (https://imagej.net/imagej-wiki-static/Sholl) in Fiji ImageJ. This method assesses signal complexity and spatial distribution relative to the organoid's center. A thresholded image of the organoid section was used, and a straight line of uniform thickness was manually drawn across the widest diameter of the section, determining the geometric center. The plugin then generated a series of concentric circles, centred on this point, at predefined radial increments. Each circle represents a distance bin for quantifying COL4A1 signal density. The software quantified the number of COL4A1-positive signals intersecting each concentric circle, starting from the centre and moving radially outward toward the periphery of the organoid section. This process was repeated across multiple organoid sections to ensure robust sampling. Accordingly, a Sholl profile was produced and exported as a distribution table. The data were further analysed statistically and visualised in GraphPad Prism, where a nonlinear regression curve was fitted to the distribution to model spatial signal decay or clustering patterns.

### Proteomics and western blot

**Sample preparation.** Organoids were placed on ice and washed twice with 1× PBS for 5 min. In the third wash, organoids were placed in 100 mM Tris-HCL, pH = 7.5. Proteins were extracted using a lysis buffer (5% SDS 100 mM Tris pH = 7.5). The lysates were transferred into soft tissue homogenizing CK14 tubes (Bertin Corp), placed in a homogenizer shaker at 400 bps for 2 sec, and then put on ice for 20 min. Samples were centrifuged at 12,000 × *g* at 4 °C for 20 min and then placed on ice. The uppermost, transparent liquid was transferred to a new Eppendorf tube and sent immediately to the De Botton Protein Profiling Institute of the Nancy and Stephen Grand Israel National Centre for Personalised Medicine, Weizmann Institute of Science, for a 'global quantification of protein' mass spectrometry. This discovery experiment aimed to quantify as many proteins as possible using label-free methods. Samples included control and *LIS1*[+/−] mutated hippOs and corticOs on days 70 and 105, respectively. Leftovers were aliquoted and used for Western blot verifications.

Cell lysates were subjected to in-solution tryptic digestion using the suspension trapping (S-trap) method as previously described[92]. Briefly, lysates were incubated at 96 °C for 5 min, followed by six cycles of 30 s of sonication (Bioruptor Pico, Diagenode, USA). Protein concentration was measured using the BCA assay (Thermo Scientific, USA). From each sample 20 μg of total protein were reduced with 5 mM dithiothreitol and alkylated with 10 mM iodoacetamide in the dark. Each sample was loaded onto S-Trap microcolumns (Protifi, USA) according to the manufacturer's instructions. After loading, samples were washed with 90:10% methanol/50 mM ammonium bicarbonate. Samples were then digested with trypsin (1:50 trypsin:protein ratio) for 1.5 h at 47 °C. The digested peptides were eluted using 50 mM

ammonium bicarbonate. Trypsin (1:50 trypsin:protein ratio) was added to this fraction and incubated overnight at 37 °C. Two more elutions were made using 0.2% formic acid and 0.2% formic acid in 50% acetonitrile. The three elutions were pooled together and vacuum-centrifuged to dryness. Samples were resuspended in $H_2O$ with 0.1% formic acid and subjected to solid phase extraction (Oasis HLB, Waters, Milford, MA, USA) according to manufacturer instructions and vacuum-centrifuged to dryness. Samples were kept at −80 °C until further analysis.

**Liquid chromatography.** ULC/MS grade solvents were used for all chromatographic steps. Dry digested peptides were dissolved in 97:3% $H_2O$/acetonitrile + 0.1% formic acid. Each sample was loaded using split-less nano-Ultra Performance Liquid Chromatography (nanoElute Bruker Daltonics, Germany). The peptides were separated using an Aurora column (75 µm ID × 25 cm, IonOpticks) at 0.3 µL/min. Peptides were eluted from the column into the mass spectrometer using the following gradient: 2% to 27% B in 100, then back to initial conditions.

Each sample was loaded using split-less nano-Ultra Performance Liquid Chromatography (10 kpsi nanoAcquity; Waters, Milford, MA, USA). The mobile phase was: A) $H_2O$ + 0.1% formic acid and B) acetonitrile + 0.1% formic acid. The samples were desalted online using a reversed-phase Symmetry C18 trapping column (180 µm internal diameter, 20 mm length, 5 µm particle size; Waters). The peptides were then separated using a T3 HSS nano-column (75 µm internal diameter, 250 mm length, 1.8 µm particle size; Waters) at 0.35 µL/min. Peptides were eluted from the column into the mass spectrometer using the following gradient: 4% to 33% B in 155 min, 33% to 90% B in 5 min, maintained at 90% for 5 min and then back to initial conditions.

**Mass Spectrometry.** The nanoUPLC was coupled online to a Time-of-flight mass spectrometer (timsTOF Pro, Bruker, Daltonics, Germany). Data was acquired in data-dependent acquisition with ion mobility mode (data-dependent acquisition (DDA) PASEF[93] using a 1.1 sec cycle-time method with 10 MS/MS scans. For ion mobility 1/K0 range was 0.60–1.60 Vs/cm², Energy Start in PASEF CID was set to 20.0 eV, and Energy End was set to 59.0 eV. Other parameters were kept as the default parameters of the DDA PASEF method.

For hippocampal organoids proteomics: The nanoUPLC was coupled online through a nanoESI emitter (10 µm tip; New Objective; Woburn, MA, USA) to a quadrupole orbitrap mass spectrometer (Q Exactive HF, Thermo Scientific) using a FlexIon nanospray apparatus (Proxeon).

Data was acquired in DDA mode, using a Top10 method. MS1 resolution was set to 120,000 (at 200 m/z), mass range of 375-1650 m/z, AGC of 3e6 and maximum injection time was set to 60 msec. MS2 resolution was set to 15,000, quadrupole isolation 1.7 m/z, AGC of 1e5, dynamic exclusion of 45 sec and maximum injection time of 60 msec.

**Data processing and analysis.** The raw data was processed with FragPipe v17.1. The data was searched with the MSFragger search engine v3.4 against the human (*Homo sapiens*) protein database as downloaded from Uniprot.org, appended with common lab protein contaminants. Enzyme specificity was set to trypsin, and up to two missed cleavages were allowed. Fixed modification was set to carbamidomethylation of cysteines, and variable modification was set to oxidation of methionines and protein N-terminal acetylation. The quantitative comparison was calculated using Perseus v1.6.0.7. Decoy hits were filtered out, and only proteins that had at least two valid values after logarithmic transformation in at least one experimental group were kept. For statistical calculations, missing values were replaced by random values from a normal distribution using the Imputation option in Perseus (width 0.3, downshift 1.8). A Student's *t* test, after the logarithmic transformation, was used to identify significant differences between the experimental groups across the

biological replica. Fold changes were calculated based on the ratio of geometric means of the different experimental groups.

Raw data was processed with MaxQuant v1.6.6.0[94]. The data was searched with the Andromeda search engine against the human (*Homo sapiens*) protein database as downloaded from Uniprot (www.uniprot.com), and appended with common lab protein contaminants. Enzyme specificity was set to trypsin, and up to two missed cleavages were allowed. Fixed modification was set to carbamidomethylation of cysteines, and variable modifications were set to oxidation of methionines, asparagine and glutamine deamidation, and protein N-terminal acetylation. Peptide precursor ions were searched with a maximum mass deviation of 4.5 ppm and fragment ions with a maximum mass deviation of 20 ppm. Peptide and protein identifications were filtered at an FDR of 1% using the decoy database strategy (MaxQuant's "Revert" module). The minimal peptide length was 7 amino acids, and the minimum Andromeda score for modified peptides was 40. Peptide identifications were propagated across samples using the match-between-runs option checked. Searches were performed with the label-free quantification option selected. The quantitative comparisons were calculated using Perseus v1.6.0.7. Decoy hits were filtered out, and only proteins that were detected in at least two replicates of at least one experimental group were kept. For statistical calculations, missing values were replaced by random values from a normal distribution using the Imputation option in Perseus (width 0.3, downshift 1.8). A Student's *t* test, after logarithmic transformation, was used to identify significant differences between the experimental groups across the biological replica. Fold changes were calculated based on the ratio of geometric means of the different experimental groups. Additionally, raw data was searched for PTMs using the GPTMD module of the MetaMorpheus v0.0.311 algorithm[95]. The data were searched against the same database described above.

## Micropipette aspiration

MPA was performed using a manometer setup as reported previously[39]. The organoids were placed, immersed in media, on a glass coverslip mounted on top of an inverted fluorescence microscope stage (Nikon Eclipse, Ti-U). The pipettes (Pipette brand and model) were washed in media with serum to lubricate the inner surface walls and decrease friction. Creep tests were performed over several seconds under 0.5-to-3 kPa suction-generated load consistent with the microenvironmental elasticity of healthy and sclerotic brain tissues.

For each organoid, the pipette inlet was brought into contact and creep test was performed by applying 1-to-3 kPa constant suction, which was measured relative to atmospheric pressure via a pressure transducer (Validyne, Northridge CA, USA). Time-lapse imaging of the aspiration dynamics into the pipette was recorded over ~ten sec and included ~two sec before applying suction. We used pipettes with 0.15-to-0.5 mm inner radii to measure tissue-level mechanics, thus probing integrated multicellular and ECM contributions. Specifically, inner diameter 0.3 mm (B100-30-7.5HP, SUTTER INSTRUMENTS) was used on Day-9 and Day-18 corticOs, 0.5 mm (B100-50-7.5HP, SUTTER INSTRUMENTS) for Day-35, and 1 mm (BF200-100-10, SUTTER INSTRUMENTS) for 70-day-old coricOs.

This creep test response to applied load is characteristic of the minimal linear viscoelastic SLS model. In its Maxwell representation, it consists of an elastic element (spring $k_1$) that is connected in parallel to a second elastic element (spring $k_2$) positioned in series with a viscous element (dashpot μ) (Fig. 1a'). 

The effective time-dependent deformability of the organoids is given by the creep compliance function $J$(t), which we obtain over a range of small deformations using the half-space model relationship[96]:

$$J(t) = \frac{2\pi L(t)}{3\phi RpP} \tag{1}$$

The length of the aspirated fraction ($L(t)$) is scaled by the pipette inner radius. $RpP$ is the applied pressure (relative to atmospheric pressure), and $\varnothing = 2.1$ is a geometrical pipette wall factor. The organoids exhibited a creep compliance function that was consistent with the SLS model, allowing us to quantitatively evaluate the three viscoelastic parameters by fitting[97]:

$$J_{SLS}(t) = \frac{1}{k_{st}}\left(1 - \frac{k_0 - k_{st}}{k_0}e^{-\frac{t}{\tau}}\right) \qquad (2)$$

In this physically intuitive representation, the instantaneous stiffness $k_O = k_1 + k_2$ measures the elastic resistance to abrupt impact. The steady-state stiffness that determines the long-term restoring forces is $k_{st} = k_1$. The time scale for the transition from elastic stretching to steady-state deformation is estimated by:

$$\tau = \frac{\mu(k_1 + k_2)}{k_1 k_2} \qquad (3)$$

Satisfyingly, the R-square goodness of fit of all organoid measurements was high (typically > 0.98), confirming the choice of the minimal linear viscoelastic SLS model (Supplementary Fig. 2b–f). Consistently, the goodness of fit of the mean creep compliance functions averaged across multiple organoids per condition was > 0.99 (Fig. 1b–e).

**Blebbistatin treatment.** The engaging pressure of the pipette with the organoids was about 1-1.5 kPa. Then, we increased the suction pressure to start aspiration and continued to follow the creep length until it reached the steady state. For control, the suction pressure required for aspiration was 1.5 kPa on average, and for mutated organoids, the average pressure required to initiate aspiration was 2 kPa. Organoids were treated with Myosin ATPase inhibitor blebbistatin (Racemic Blebbistatin, Sigma Aldrich Catalog no. 203389) at a concentration of 50 µM and were incubated in the incubator for 2 hr. For the blebbistatin-treated mutated organoids, we used an average of 1.2 kPa. A pipette of 25-micron in diameter was used for all three conditions.

### ELISA activity assay

The MMP9 catalytic domains were incubated in a 96-well plate with different concentrations of Mca-PLGL-Dpa-AR-NH2 Fluorogenic MMP Substrate (ES001, R&D SYSTEMS), a substrate that is cleaved by MMP9. The peptide substrate contains a highly fluorescent 7-methoxy coumarin group that is quenched when cleaved. Active peptidases that can cleave an amide bond between the fluorescent group and the quencher group cause an increase in fluorescence, which can confirm their function[98]. Each well was supplemented with 20 nM of either MMP9, a TNC digestion buffer (50 mM TRIS-HCl pH = 7.5, 150 mM NaCl, 5 mM CaCl$_2$, and 0.05% Brij), and the substrate at a dilution series (1–100 µM). Two control conditions were 'TNC-only' and a 10 µM 'TNC + substrate' condition. While at 37 °C, fluorescence was captured by a plate reader (Synergy H1) every 50 seconds for an overall duration of 40 minutes (Supplementary Fig. 5).

### MMP9 catalytic domain treatment

On day 18, control and $LIS1^{+/-}$ organoids were treated with 500 nM of MMP9 catalytic domain diluted in the organoids' original media and incubated for 10 min at 37 °C, 5% CO$_2$. The treated media was then replaced with fresh 100 µl of 'SA1 media', and plates were held in the incubator until the organoids were tested.

### Diffusion MRI–data analysis

The diffusion-weighted imaging (DWI) MRI dataset was collected to assess the differences between brain organoid groups. Based on the collected dataset, the ADC maps were calculated using a mono-

exponential fit. The images with the highest b-value (the degree of diffusion weighting) were used to segment and identify the voxels of the brain organoid tissue. Supplementary Fig. 5a shows the ADC maps and the contour of the segmented voxels. A normalized distribution of the ADC values in the identified voxels for each type, consisting of 100 bins, was defined. A denoising of the distribution profile was then performed, removing high-frequency components using FT. The distribution of the diffusion in the brain organoid tissue results in an asymmetric profile; therefore, the maximal likelihood position was defined as the center of points with 2/3I$_{max}$. intensity (Imax was found based on the denoised distribution, and the two points had 2/3I$_{max}$ intensity from both sides of the distribution by interpolation). The same steps were repeated for each organoid type and scan. Supplementary Fig. 5a–c shows the ADC values distributions and the estimated maximal likelihood positions. Note that the ADC values increased as a function of the repeated scans. This can be due to the long scan duration during which the organoids were out of their regular environment. However, the deviation of LIS1$^{+/-}$ compared to Control was preserved over the scan duration. Three additional experiments were performed with different subsets of the organoid types. Supplementary Fig. 5d shows a similar observed trend in the repeated experiments. The ages of the organoids in the different experiments were: experiment 1, 19 days; experiment 2 & 3, 18 days; experiment 4, 16 days. The lower ADC values in Exp.#4 can be due to different scan parameters (with a slice thickness of 100 µ compared to 200 µ; see the scan parameters summarized below.

MRI scanner details: Horizontal Bruker Biospec 15.2-T USR preclinical MRI scanner with an Avance IIIHD console.

Container used for the MRI scans: In experiments #1–3, the organoids were placed in a container with separated wells for each type, allowing for scanning all types in one scan. Exp. #4 was scanned with a single Eppendorf in which the organoids of one kind were placed and scanned, and then the scan was repeated with the second type. The medium used was PBS.

RF coil details: 1H MRI CryoProbe 2-element array kit for the mouse head.

The DWI scan parameters:
- Exp.#1: TR/TE 400/13.7 ms, FOV 12 × 12 mm$^2$, in-plane resolution 100 × 100 µ$^2$, slice thickness 200 µ, 20 slices, ten averages, scan duration 48 minutes. The scan was repeated five times. The scan included six b-values - 0, 200, 400, 600, 1000, 1200 s/mm$^2$.
- Exp. #2: TR/TE 400/13.7 ms, FOV 18 × 12 mm$^2$, in-plane resolution 100 × 100 µ$^2$, slice thickness 200 µ, nine slices, ten averages, scan duration 48 minutes. The scan included six b-values - 0, 200, 400, 600, 1000, 1200 s/mm$^2$.
- Exp.#3: TR/TE 400/13.7 ms, FOV 12 × 12 mm$^2$, in-plane resolution 100 × 100 µ$^2$, slice thickness 200 µ, 20 slices, 20 averages, scan duration 1h36m0s0ms. The scan included six b-values - 0, 200, 400, 600, 1000, 1200 s/mm$^2$.
- Exp.#4: TR/TE 300/12.7 ms, FOV 12.6 × 2.4 mm$^2$, in-plane resolution 100 × 100 µ$^2$, slice thickness 100 µ, 60 slices (3D acquisition), one average, scan duration 1h 12m 0s 0ms. The scan included four b-values–0, 200,600,1000 s/mm$^2$.

### RNA and miR sequencing

Total RNA from 105 days old corticOs ($N = 4$; $n = 6$–8 per genotype) and from 18 days old MMP9-treated and untreated organoids ($N = 4$, of which two outlier groups from the non-treated LIS1 +/− samples were excluded from the analysis; $n = 12$–14) was extracted using the Directzol RNA Miniprep Plus kit (Zymo Research) following the manufacturer's protocols. RNA concentration and integrity were measured using Nanodrop (Thermo Scientific). Samples were then sent to the DNA Link Sequencing Lab in S. Korea, where RNA-seq libraries were prepared using TrueSeq standard mRNA library kit or small RNA-seq libraries. Libraries were sequenced in NovaSeq6000. The raw data was

processed using the Weizmann Institute bioinformatics pipeline. DE genes were analyzed using Metascape[42] and GeneAnalytics[44].

## Computational model

The energy functional for the mechanics of the microstructure of a brain organoid with N nodes and $N_C$ number of cells:

$$E = \frac{k_s}{2} \sum_{<ij>} p_{ij} \left( l_{ij} - l_0 \right)^2 + \frac{k_b}{2} \sum_{<ijk>} p_{ij} p_{jk} \left( \theta_{ijk} - \pi \right)^2 + \sum_{n=1}^{N_C} \lambda_n (A_n - A_0)^2 \tag{4}$$

Here, $k_s$ denotes the individual ECM fiber stretching stiffness, while $k_b$ denotes the individual ECM fiber bending stiffness to give a persistence length. Moreover, $p_{ij}$ represents the edge occupation probability, or the fraction of edges in the network that are occupied. We set $p_{ij} = p$ for convenience in the main body of the manuscript. As $l_{ij}$ quantifies an edge length in the network, the first term in the equation above captures the stretching energy. As $\theta_{ijk}$ denotes the angle between two edges, the second sum in the equation above quantifies the semiflexibility of the fiber network, while the third term encodes the bulk stiffness of $N_c$ cells in each area $A_n$ and stiffness $\lambda_n = \lambda q_n$, with $q_n$ either zero or one as the number of cells is set and then the triangles in the lattice are randomly occupied with cells. Therefore, λ quantifies the individual bulk cell stiffness, which is, for simplicity, the same for all cells. Note that $k_s/(k_b l_0^2) = R$ and is the dimensionless ratio comparing the individual ECM fiber stretching stiffness to bending stiffness.

To measure the stiffness K of this composite system, once the model parameters are chosen, we impose periodic boundaries in the horizontal direction and cut, delete the periodic edges in the vertical direction, and then compress the system in the vertical direction by applying strain γ. We then compute the energy E of the system, the first derivative of the energy to obtain the strain, and, finally, the second derivative of the energy of the system to compute the material stiffness K, which includes a $1/A_O$ prefactor. We make such measurements for different amounts of randomly positioned ECM (mutant) as well as for patterned ECM (control) via the removal of ECM from localized circular regions near the center.

## Reporting summary

Further information on research design is available in the Nature Portfolio Reporting Summary linked to this article.

## Data availability

The mass spectrometry proteomics data generated in this study have been deposited in the ProteomeXchange Consortium[99] via the PRIDE[100] partner repository with the dataset identifiers PXD040813 (cortical organoids), PXD040911 (hippocampal organoids), and PXD049178 (early organoids). The processed RNA-seq data generated in this study have been deposited in the GEO database under accession code GSE228926. Source data are provided with this paper.

## Code availability

The computational modeling code related to this manuscript is deposited in the Zenodo repository and available for public access at: https://doi.org/10.5281/zenodo.15028450.

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

## Acknowledgements

We express our gratitude for the help of Arpan Parichha, Alfredo Isaac Ponce Arias, Mahesh Gandikota, and Tao Zhang. Orly Reiner is an incumbent of the Berstein-Mason professorial chair of Neurochemistry and the Head of the M. Judith Ruth Institute for Preclinical Brain Research. I.S. is an incumbent of the Maurizio Pontecorvo Professorial Chair. Our research has been supported by research grants from Ethel Lena Levy, the Selsky Memory Research Project, the Gladys Monroy and Larry Marks Center for Brain Disorders, the Advantage Trust, the Nella and Leon Benoziyo Center for Neurological Diseases, the David and Fela Shapell Family Center for Genetic Disorders Research, the Abish-Frenkel RNA center, the Andrea L. and Lawrence A. Wolfe Family Center for Research on Neuroimmunology and Neuromodulation, Weizmann - Center for Research on Neurodegeneration, the Brenden- Mann Women's Innovation Impact Fund, The Irving B. Harris Fund for New Directions in Brain Research, the Irving Bieber, M.D. and Toby Bieber, M.D. Memorial Research Fund, The Leff Family, Barbara & Roberto Kaminitz, Sergio & Sônia Lozinsky, Debbie Koren, Jack and Lenore Lowenthal, and the Dears Foundation. A research grant from the Estates of Ethel H. Smith, Gerald Alexander, Mr. and Mrs. George Zbeda, David A. Fishstrom, Norman Fidelman, Hermine Miller, Olga Klein Astrachan, Hermine Miller, and The Maurice and Vivienne Wohl Biology Endowment, a research grant from Emily Merjan, the ISF grant (545/21), and the United States-Israel Binational Science Foundation (BSF; Grant No. 2023009), the support (O.R.). This research was supported by the Azrieli Institute for Brain and Neural Sciences (O.R., R.S., and I.S.). The

Alzheimer's Association Grant AARG-NTF-21-849529 supported the development of the MRI procedure (R.S.). Funding provided by BIRAX-Ageing supported the design and operation of the MPA apparatus (A.B.). Financial support for the computational studies was provided by the National Science Foundation via DMR-2204312 (J.M.S.).

## Author contributions

O.R., A.B., J.M.S., R.S., I.S., and M.K.Z. conceptualized the study. M.K.Z. conceived, conducted, and analyzed the experiments with B.B., I.Sa., S.B.D. M.K.Z. and B.B. grew the organoids for RNA-seq, proteomics, and physical measurements and conducted immunostainings. A.S., and Y.L.A.P. planned, conducted, and analyzed proteomics data. T.S., T.O., and I.Sa. contributed to data analysis and provided guidance. T.O. curated and analyzed RNA-seq and small RNA-seq data. T.H. and R.S. planned, conducted, and analyzed MRI experiments. I.S. provided experimental planning and guidance. J.M.S. led computational modeling. A.B. conducted and analyzed micropipette aspiration experiments. O.R. supervised the project and secured funding. I.Sa. co-mentored M.K.Z. and secured funding. O.R. wrote the manuscript with input from all authors, M.K.Z. and B.B. generated all the figures. All of the authors reviewed and edited the manuscript. All authors approved the final version of the manuscript.

## Competing interests

The authors declare no competing interests.
