## [Transparent Peer Review file · Nature Communications]

Altered Extracellular Matrix Structure and Elevated Stiffness in a Brain Organoid Model for Disease

Corresponding Author: Professor Orly Reiner

Version 0:

Reviewer comments:

Reviewer #1

(Remarks to the Author)

In the manuscript by Karlinsky-Zur et al, the authors investigated the mechanical properties of brain organoids mutant for LIS1, a gene that mutated causes lissencephaly. I think this is potentially an interesting paper. It is intriguing to correlate the functionality of the ECM in a disease organoid model. I do have, however, some comments that should be carefully considered by the authors:

-In the paper the authors do not specify if they use Matrigel in the brain organoid generation protocol, but they refer to their previous paper in which they state Matrigel is used. Maybe I missed this information. Since they investigate the ECM compositions of their organoids, I think some validations of their key results (especially the matrisome characterization) in organoids generated Matrigel free (e.g using the Pasca spheroid protocol or the recent Martins-Costa, EMBO J) is needed.

- I think it is important to use the proper control brain organoids. The methods are written quite generally so it is not very easy to find this information, but it is important that to generate control organoids unedited ES cells that underwent the same procedure of the edited LIS1 cells (i.e. same electroporation with an empty gRNA plasmid and Cas9, FACS procedure etc etc.). It is very well known that any subtle difference in ES state can greatly influence the resulting organoids and this is why simply using unedited= not electroporated ES cannot be considered a proper control.

-The authors compare organoids stiffness at days 9, 18,35,70 and they see differences in stiffness between wt and mt organoids (Fig. 1). However, they then decide to do mass spec and most of the other analysis at day 105: is the difference of stiffness still present at that day- or viceversa, are there differences in the mass spec data of the previous time points? I think this is a quite important missing data.

-Treatment with MMP9 partially rescued the stiffness phenotype as well as gene expression. Do the authors also observe the cellular differences being rescued (e.g. data showed in Fig. SF8 and SF9)? For example: can a repeated MMP9 treatment influence progenitor cells number?

-It is important that for every quantification/experiment the author could report how many organoids per batch/number of batches/number of clones analyzed etc etc. It would help the reader to understand the robustness of the conclusions.

-Could the authors elaborate on the significance of the decrease of the gammaH2X in the LIS1 mutants and the possible correlation with LamA increase?

Minor comments:

“To our knowledge, this is the first study to elucidate how tissue mechanics change in disease states using human brain organoids”. What is the advantage and the outlook of this new study for the field? I think it is a very interesting topic to be put into perspective.

In the methods section information are generally not elaborate. Please revise.

In the intro the authors state: “Studying this disease in mouse models has been useful but limited since their cortex naturally

lacks convolutions". But also brain organoids lack convolutions.

(Remarks on code availability)

Reviewer #2

(Remarks to the Author)

In this manuscript, Zur et al. describe changes in ECM composition, mechanical properties and water content of brain organoids with LIS1 mutations. They find the LIS1 mutations correlate with an increase in organoid stiffness, and that degrading the ECM by MMP9 leads to a softening of the organoids with LIS1 mutations and changes in gene expression levels. Finally, the authors developed a computational model to capture their findings. This study continues the work by Kshirsagar et al (Nat Commun 2023), where the lab has already shown an effect of LIS1 on ECM expression and cell stiffness.

The experiments presented in the current study appear sound, they are clearly described and properly analyzed. The findings that increased collagen levels and Lamin A/C levels lead to a stiffening of cells and tissues, that digesting collagen softens tissue, and that changes in tissue stiffness are associated with changes in gene expression patterns are in agreement with a large body of existing literature. Overall, the authors report mostly correlations and the manuscript remains rather descriptive, mostly confirming previous findings in a new organoid system. As they do not explore whether MMP treatment has any effect on brain malformations, it is not clear how relevant the reported changes in organoid mechanics and ECM structure are for lissencephaly.

Specific points:

1) The authors should show whether MMP treatment leads to changes in Lamin A/C levels, and if not they should discuss why organoid stiffness drops to control levels and does not remain stiffer than controls.

2) I'm surprised that a 10 min treatment with MMP9 was sufficient to see an effect on organoid mechanics. The authors should discuss how quickly MMP9 diffuses into the organoid (perhaps estimate how deep into the organoid it may diffuse at all) and how quickly it may digest the ECM. Could this perhaps be the result of an active response of the cells to the enzyme? For the estimation of diffusion depth, an experiment with fluorescently labelled markers of similar size would be helpful. A control experiment in which cellular contractility is perturbed (e.g. using blebbistatin) would allow distinguishing between cell and ECM effects on the measured organoid mechanics.

3) Lissencephaly may be caused by LIS1 mutations as stated in the abstract but doesn't have to be. I'd recommend rephrasing that sentence.

(Remarks on code availability)

Version 1:

Reviewer comments:

Reviewer #1

(Remarks to the Author)

We thank the authors for addressing most of our suggestions. Especially, the addition of the requested controls in my opinion increases the robustness of the results. I have no further concerns.

(Remarks on code availability)

Reviewer #2

(Remarks to the Author)

While the authors have partially addressed some of my minor concerns (see below), unfortunately, they have not addressed my main concern, the lack of novelty, at all. The study by Zur et al. remains largely descriptive and mostly confirms previously published results in their organoid system. The authors motivate their study by the potential importance of ECM and mechanics in lissencephaly, but provide no data on effects on brain folding or on mechanisms linking the scaffold protein LIS1 to tissue-level mechanics. They only show correlations between LIS1 mutations and organoid mechanics, ECM composition and water content. As mentioned above, the lab has already published that LIS1 expression affects ECM and cell stiffness (Kshirsagar et al 2023). That digestion of ECM alters tissue mechanics has been published many times, including in in vivo models (e.g. Bansaccal, Nature 2023), as has the regulation of stem cell function by matrix stiffness and ECM (e.g. Jiang et al, NatComms 2024). In their abstract, the authors emphasize that 'this is the first study to elucidate how tissue mechanics change in disease states using human brain organoids'. While this may well be the case, there have been

many published in vivo studies that elucidate how brain tissue mechanics change in disease states using living humans (largely studies using magnetic resonance elastography). Without providing insights into potential disease mechanisms, progress therefore remains rather limited.

Regarding my minor comments: the authors did not show that the ECM is significantly altered after 10 minutes of MMP9 treatment; it is not clear how the references they provided are relevant to answer this question. I am not convinced that the softening they see is due to permanent changes in the ECM.

(Remarks on code availability)

Dear Reviewers,

We extend our sincere appreciation to the anonymous reviewers for their insightful comments and constructive feedback, which have significantly enhanced the quality of this manuscript.

Reviewer #1

In the manuscript by Karlinsky-Zur et al, the authors investigated the mechanical properties of brain organoids mutant for LIS1, a gene that mutated causes lissencephaly. I think this is potentially an interesting paper. It is intriguing to correlate the functionality of the ECM in a disease organoid model. I do have, however, some comments that should be carefully considered by the authors:

- 1. In the paper the authors do not specify if they use Matrigel in the brain organoid generation protocol, but they refer to their previous paper in which they state Matrigel is used. Maybe I missed this information. Since they investigate the ECM compositions of their organoids, I think some validations of their key results (especially the matrisome characterization) in organoids generated Matrigel free (e.g using the Pasca spheroid protocol or the recent Martins-Costa, EMBO J) is needed.**

We did not use Matrigel, and revised the methods section to clarify that. The modified text was copied in response to the “minor comment B” concerning the methods section.

- 2. I think it is important to use the proper control brain organoids. The methods are written quite generally so it is not very easy to find this information, but it is important that to generate control organoids unedited ES cells that underwent the same procedure of the edited LIS1 cells (i.e. same electroporation with an empty gRNA plasmid and Cas9, FACS procedure etc etc..). It is very well known that any subtle difference in ES state can greatly influence the resulting organoids and this is why simply using unedited= not electroporated ES cannot be considered a proper control.**

We generated two additional control lines evaluated through proteomics and biomechanical analyses.

PX335 Control: The pX335 plasmid, an empty Cas9 nickase plasmid used in creating the original LIS1 mutant, was electroporated into the parental WIBR3 line to produce this new control line. Organoids were generated from this line in parallel with those from the original WIBR3 line and were harvested at day 35 for proteomics analysis. While the pX335 control line showed some differentially expressed proteins compared to the original control line, both control lines exhibited significant differences when compared to

the LIS1 mutant lines. A prominent extracellular matrix signature was observed in the proteomics data (Fig. 2a).

C20 Control: A single colony was isolated from the WIBR3 line through subcloning and used as a control for rheology experiments, as shown in Supp. Fig. 5. Consistent with previous findings, organoids derived from LIS1 mutants were stiffer than those derived from the control line.

- 3. The authors compare organoids stiffness at days 9, 18, 35, 70 and they see differences in stiffness between wt and mt organoids (Fig. 1). However, they then decide to do mass spec and most of the other analysis at day 105: is the difference of stiffness still present at that day- or viceversa, are there differences in the mass spec data of the previous time points? I think this is a quite important missing data.**

In the paper, proteomics analysis was conducted on 18-day-old organoids (ectoderm), 70-day-old hippocampal organoids, and 105-day-old cortical organoids. Following your comment, we conducted an additional mass-spectrometry on day 35, this time using a new control - the PX335 control, along with the WIBR3 control and LIS1 mutant lines used previously. Results from the metascape analysis are now presented in Fig. 2a, showing that both controls differ mostly from the mutated line in proteins associated with the extracellular matrix. The reads database can be found in Supplementary Table S1a, and the full results of the metascape p-values are shown in Supplementary Table S1b.

Text was modified as follows: “To delineate the molecular changes that are associated with the stiffening of the *LIS1*^{+/-} cortices, we analyzed the proteomic signature of control and *LIS1*^{+/-} corticOs. On day 35, we extracted proteins from the *LIS1*^{+/-} mutated corticOs, WIBR3 control and a PX335 control. The PX335 control line was created with the empty Cas9 nickase plasmid used in creating the original LIS1 mutant,

electroporated into the parental WIBR3 line to produce a second control CRISPR CorticOs. A total of 7,842 proteins were identified and quantified (**Supplementary Table S1a**), of which 274 were DE in the mutation, based on the threshold criteria of Log2Fold change $\geq |1|$, at least 1 peptide per protein, and ANOVA p-value < 0.05 in both PX335 and WIBR3 control vs. *LIS1*^{+/-} comparisons. We then conducted a Metascape analysis to examine the pathways differing between the two control lines and the mutated corticOs, and identified proteins associated with the extracellular matrix as most affected (**Fig.2a**)”

- 4. Treatment with MMP9 partially rescued the stiffness phenotype as well as gene expression. Do the authors also observe the cellular differences being rescued (e.g. data shown in Fig. SF8 and SF9)? For example: can a repeated MMP9 treatment influence progenitor cell number?**

We did not check progenitors since that would require a longer treatment than 10 minutes. We noted that a long-term treatment had complicated effects in the past, and interpreting the results has been proven difficult. We checked the immediate effect on gene expression as detailed in the manuscript (Fig. 4).

- 5. It is important that for every quantification/experiment the author could report how many organoids per batch/number of batches/number of clones analyzed etc etc. It would help the reader to understand the robustness of the conclusions.**

Thank you for your comment. We clarified and / or added missing values in the following figures and supplementary figures:

- Fig. 2: Matrisome composition of CorticO. The analysis included: **(a)** Day-35, for each genotype $N_{corticOs} = 4$, $n_{corticOs} = 10-12$, **(b-c)** Day-105 proteomics and WB: $N_{corticOs} = 4$, $n_{corticOs} = 6-8$.
- Fig. 3: The effects of *LIS1*^{+/-} mutation and ECM proteolysis on CorticOs mechanics and structural organization. Symbols and error bars correspond to mean and SEM across $N_{WT} = 4$, $N_{10F} = 5$, $N_{WT-MMP9} = 5$, $N_{10F-MMP9} = 4$ organoids.
- Fig. 4: Rescued genes and inverse interaction between mRNA and miRNA in *LIS1*^{+/-} organoids. **a**, A heatmap showing the top rescued genes and their expression before and after the MMP9 treatment, $N = 4^*$, $n = 12-14$.
- * Two outlier groups from the non-treated *LIS1*^{+/-} samples were excluded from the analysis
- Fig. 5: Collagen organization and a microstructure mechanical model. The Sholl analysis included: Day 9 - $N_{WT} = 2$, $N_{LIS1+/-} = 3$; Day 18 - $N_{WT} = 3$, $N_{LIS1+/-} = 3$.

Likewise, we now also ensured that all the number of organoids are specified also in the supplementary figures:

- Supplementary Fig. S2: Creep test MPA rheology. **Day-9**, $N_{WT} = 6$, $N_{10F} = 8$, $N_{9G} = 7$. **Day-18**, $N_{WT} = 4$, $N_{10F} = 5$, **Day-35**, $N_{WT} = 8$, $N_{10F} = 8$, and **Day-70**, $N_{WT} = 7$, $N_{10F} = 9$.
- Supplementary Fig. S3: RNA-seq of 105 days old corticOs reveal altered ECM expression. The analysis included: $N_{corticOs} = 4$; $n_{corticOs} = 6-8$.
- Supplementary Fig. S4: Hippocampal organoids characterization and comparative proteomics of LIS1^{+/-} and control samples. The mass spectrometry included $N_{hippOs} = 3$; $n_{hippOs} = 5-8$.
- Supplementary Fig. S7: miRNA-seq of 105 days old corticOs. $N_{corticOs} = 4$; $n_{corticOs} = 6-8$.

6. Could the authors elaborate on the significance of the decrease of the gammaH2X in the LIS1 mutants and the possible correlation with LamA increase?

Response: We added to the text: “The decrease in γ H2AX levels in LIS1 mutants is significant, as γ H2AX marks DNA damage and recruits repair machinery to maintain genomic stability. Reduced γ H2AX suggests impaired DNA damage response (DDR), potentially hindering effective repair and leading to mutation accumulation. This may exacerbate developmental defects in highly dynamic tissues like the developing brain.”

Minor comments:

- A. “To our knowledge, this is the first study to elucidate how tissue mechanics change in disease states using human brain organoids”. What is the advantage and the outlook of this new study for the field? I think it is a very interesting topic to be put into perspective.**

Response: We thank the reviewer for this comment and elaborate on it below. We modified the discussion accordingly. “Our study reveals the critical role of tissue mechanics in brain development, demonstrating how viscoelastic properties are altered in lissencephaly caused by LIS1 mutations. Using human brain organoids, we uncovered key mechanical and molecular changes, such as increased stiffness and abnormal ECM organization, providing insights into previously understudied aspects of the disease. Combining rheological measurements, MRI imaging, molecular analyses, and computational modeling, we showed that short-term MMP9 treatment can reverse stiffness and water diffusion abnormalities, highlighting the therapeutic potential of targeting ECM dysregulation. These findings establish tissue mechanics as a vital factor in brain malformations and a promising target for intervention. Our research sets a new paradigm for studying mechanical principles in brain disorders, advancing mechanobiology in neurodevelopmental and neurological diseases while offering innovative therapeutic targets.”

- B. In the methods section information is generally not elaborate. Please revise.**

The method sections were modified as follows:

- Cell lines. The pX335 plasmid, an empty Cas9 nickase plasmid used in creating the original *LISI*^{+/-} cell line, was electroporated into the parental WIBR3 line to produce a second control. The PX335 line was used for the proteomic analysis on 30 days old organoids. Finally, a single colony was isolated from the WIBR3 line through subcloning and used as an additional control for the rheology experiment on day 35.
- Generation of hESC-derived corticOs and hippOs. Organoids were not embedded or exposed to Matrigel.
- Sholl analysis quantification (attached the overall modified text). The distribution of COL4A1 signal in day 9 and day 18 corticOs was determined using the Sholl analysis plugin (link) in Fiji ImageJ. This method assesses signal complexity and spatial distribution relative to the organoid's center. A thresholded image of the organoid section was used, and a straight line of uniform thickness was manually drawn across the widest diameter of the section, determining the geometric center. The plugin then generated a series of concentric circles centered on this point at predefined radial increments. Each circle represents a distance bin for quantifying COL4A1 signal density. The software quantified the number of COL4A1-positive signals intersecting each concentric circle, starting from the center and moving radially outward toward the periphery of the organoid section. This process was repeated across multiple organoid sections to ensure robust sampling. Accordingly, a Sholl profile was produced and exported as a distribution table. The data were further analyzed statistically and visualized in GraphPad Prism, where a nonlinear regression curve was fitted to the distribution to model spatial signal decay or clustering patterns.
- Immunohistochemistry. Citrate buffer concentration was added.
- Image analysis and quantification. Information about microscope and lens details were added: “Immunostained slides were imaged with Andor Dragonfly spinning disk confocal microscope system HC FLUOTAR 25X/0.95 W VISIR lens and stitched to get complete images of the whole organoid sections using the Andor: Oxford Instruments Fusion Shell software.”
- Blebbistatin treatment. Engaging pressure of the pipette with the organoids was about 1-1.5 kPa. Then we increased the suction pressure to start aspiration and continued to follow the creep length until it reached the steady state. For control the suction pressure required for aspiration was 1.5 kPa on average, and for mutated organoids the average pressure required to initiate aspiration was 2 kPa. Organoids were treated with Myosin ATPase inhibitor blebbistatin (Racemic Blebbistatin, Sigma Aldrich Catalog no. 203389) at a concentration

of 50 μM and were incubated in the incubator for 2 hr. For the blebbistatin treated mutated organoids we used an average of 1.2 kPa. A pipette of 25-micron in diameter was used for all three conditions.

C. In the intro the authors state: “Studying this disease in mouse models has been useful but limited since their cortex naturally lacks convolutions”. But also brain organoids lack convolutions.

We agree with this statement but want to indicate that human cortical organoids have the molecular potential to fold. We will modify the text accordingly. Modified text: “Studying this disease in mouse models has been useful but limited since their cortex naturally lacks convolutions, and human brain organoids, which usually lack folds, have the potential to form them, as we have previously demonstrated ¹.”

1. Karzbrun, E., Kshirsagar, A., Cohen, S.R., Hanna, J.H. & Reiner, O. Human Brain Organoids on a Chip Reveal the Physics of Folding. *Nat Phys* **14**, 515-522 (2018).

Reviewer #2 (Remarks to the Author)

In this manuscript, Zur et al. describe changes in ECM composition, mechanical properties and water content of brain organoids with LIS1 mutations. They find the LIS1 mutations correlate with an increase in organoid stiffness, and that degrading the ECM by MMP9 leads to a softening of the organoids with LIS1 mutations and changes in gene expression levels. Finally, the authors developed a computational model to capture their findings. This study continues the work by Kshirsagar et al (Nat Commun 2023), where the lab has already shown an effect of LIS1 on ECM expression and cell stiffness.

The experiments presented in the current study appear sound, they are clearly described and properly analyzed. The findings that increased collagen levels and Lamin A/C levels lead to a stiffening of cells and tissues, that digesting collagen softens tissue, and that changes in tissue stiffness are associated with changes in gene expression patterns are in agreement with a large body of existing literature. Overall, the authors report mostly correlations and the manuscript remains rather descriptive, mostly confirming previous findings in a new organoid system. As they do not explore whether MMP treatment has any effect on brain malformations, it is not clear how relevant the reported changes in organoid mechanics and ECM structure are for lissencephaly.

Specific points:

- 1. The authors should show whether MMP treatment leads to changes in Lamin A/C levels, and if not they should discuss why organoid stiffness drops to control levels and does not remain stiffer than controls.**

The MMP9 treatment was conducted on day 18. At this point, the levels of Lamin A/C are very low and cannot be detected by Western blot analysis (tested but not presented in text). These low Lamin A/C protein expression levels during early development have been noted previously². Our model takes into consideration the changes in either the organization and/or the levels of the ECM. If we have more ECM (as in the case of later days of LIS1 brain organoids), the organoids are stiffer. Cleavage of the ECM will result in a less stiff organoid. However, the effect will be much more pronounced in an organoid containing a lot of ECM than in an organoid with less ECM (as in the control case). In the case of day 18 organoids, we noted a striking difference in how the ECM is organized. The control ECM was organized in ring-like structures, whereas the ECM was disorganized in the case of LIS1 organoids. Once we cleaved the connections in the model, we were surprised to note that the effect was dramatic when the ECM was disorganized and more subtle when the ECM exhibited a structure.

2. Rober, R.A., Weber, K. & Osborn, M. Differential timing of nuclear lamin A/C expression in the various organs of the mouse embryo and the young animal: a developmental study. *Development* **105**, 365-378 (1989).

- 2. I'm surprised that a 10 min treatment with MMP9 was sufficient to see an effect on organoid mechanics. The authors should discuss how quickly MMP9 diffuses into the organoid (perhaps estimate how deep into the organoid it may diffuse at all) and how quickly it may digest the ECM.**

- (a) Could this perhaps be the result of an active response of the cells to the enzyme? For the estimation of diffusion depth, an experiment with fluorescently labelled markers of similar size would be helpful.**

Thank you for your comment. Although the reviewer inquired about the depth of catMMP-9 penetration into the organoid, we would like to emphasize that MMP-9 does not merely passively diffuse into the organoid since it actively degrades specific ECM molecules, reducing their concentration and density in organoids and affects the diffusion by this way. As was previously shown in collagen gels and articular cartilage, there is an inverse relationship between diffusion and matrix concentration and density^{1,2}.

Since it is very challenging to calculate precisely the kinetics of ECM degradation in brain organoids by catMMP9, we employed a conventional, convenient, and reproducible assay for MMPs to calculate kinetic

parameters of the enzyme. This assay is based on the hydrolysis of small fluorogenic peptides (see the details in the **Methods** section and the results presented in **Fig. S15**). The V_{max} , the maximum velocity, derived from the fluorogenic assay is $V_m=(185\pm33)$ uM/s, indicating that the enzyme may catalyze ~ 185 uM of substrate per second. Additionally, $K_m=(2\pm1)$ uM reflecting the affinity of the catMMP9 to the fluorogenic peptide. These calculated parameters were also added to **supplementary figure S5a** as follows: “The calculated kinetics parameters are: $V_m = (185\pm33)$ uM/s and $K_m = (2\pm1)$ uM.”

The significant effect of the ECM remodelers on organoids mechanics through ECM degradation is further supported by our MRI results and published observations showing that the injection of remodeling enzymes, such as collagenase and hyaluronidase, which degrade the ECM through different mechanisms, increased the diffusion of 2-MDa FITC-dextran in multicellular spheroids and xenografts³.

1. Ramanujan, S.; Pluen, A.; McKee, T. D.; Brown, E. B.; Boucher, Y.; Jain, R. K. *Biophysical journal* **2002**, 83, (3), 1650-1660.
2. Erikson, A.; Andersen, H. N.; Naess, S. N.; Sikorski, P.; Davies, C. d. L. *Biopolymers: Original Research on Biomolecules* **2008**, 89, (2), 135-143.
3. Eikenes, L.; Tufto, I.; Schnell, E. A.; Bjørkøy, A.; Davies, C. D. L. *Anticancer research* **2010**, 30, (2), 359-368.

(b) A control experiment in which cellular contractility is perturbed (e.g. using blebbistatin) would allow distinguishing between cell and ECM effects on the measured organoid mechanics.

Thank you for your comment. The main text and supplementary figure S5 were modified following the blebbistatin experiment: “In addition, it has been previously shown that LIS1 mutations cause biomechanical changes at the cellular level. So far, we found that $LIS1^{+/-}$ organoids were stiffer (k_0 and k_{st}) compared to control organoids and that this stiffness was reduced by MMP9 treatment. However, MPA tests with 12.5 μ m pipettes, in which cell-level mechanics can be examined, revealed that cell contractility interference by myosin ATPase inhibition of blebbistatin also resulted in softened k_0 and k_{st} parameters in 18 days old $LIS1^{+/-}$ organoids (**Supplementary Fig. S5b-d**). $LIS1^{+/-}$ organoids exhibit prolonged energy dissipation, likely due to increased extracellular matrix deposition. These findings confirm that the $LIS1$ mutation enhances stiffness and friction at both cell and tissue levels. combined, the MMP9 and blebbistatin treatments implicated both ECM and contractile forces involved in tissue stiffening of the $LIS1^{+/-}$ organoids.”

Supplementary Fig. S5: a, MMP9 catalytic domain ELISA activity assay and blebbistatin treatment results. a, Fluorescence signal intensity at uniform time intervals for two control conditions - ‘TNC-only’ and ‘TNC + substrate’ and experimental conditions with 20 nM of MMP9 catalytic domain and varying concentrations of substrate (from 1 μ M to 100 μ M). The calculated kinetics parameters are: $V_m = (185 \pm 33)$ uM/s and $K_m = (2 \pm 1)$ uM. **b-d, Cell-scale MPA confirms LIS1-mutation organoid stiffening. (b)** Representative images of human telomerase-positive skin fibroblasts that were treated with 50 μ M Blebbistatin for two hours and compared with non-treated cells. Cytoplasmic blebbing marks Myosin-II ATPase inhibition. Representative aspiration snapshots of Day-18 control, LIS1^{+/-}(10F) and blebbistatin-treated LIS1^{+/-}(10F) organoids. Snapshots were recorded after 4 sec under 1.3 ± 0.2 kPa suction. Pipette inner radius: 12.5 mm **(c)** Creep compliance and SLS fits of (i) control, (ii) LIS1^{+/-}, and (iii) blebbistatin-treated LIS1^{+/-} organoids represent multiple sites that were measured on nControl = 6, nLIS1^{+/-} = 6, nLIS1^{+/-}(Blebb) = 4 organoids per condition. **(d)** Fitted instantaneous stiffness, steady-state stiffness and response time are compared between conditions (One-Way ANOVA test N = 5, $\alpha = 0.05$).

3. Lissencephaly may be caused by LIS1 mutations as stated in the abstract but doesn't have to be. I'd recommend rephrasing that sentence.

The text was modified as follows: Lissencephaly is a severe malformation of cortical development caused by mutations in *LIS1* and other genes, which results in a lack of cortical convolutions.

Dear Reviewers,

We sincerely appreciate the time and effort you have dedicated to evaluating our manuscript. Your constructive feedback has been invaluable in refining our study. Below, we provide our detailed responses to the comments raised by each reviewer.

Response to Reviewer #1

We are grateful for your positive assessment of our manuscript and for acknowledging the robustness of the additional controls we incorporated. Your support strengthens our confidence in the study's findings. As you have no further concerns, we appreciate your time and consideration in reviewing our work.

Response to Reviewer #2

- 1. Novelty Concerns:** We acknowledge the reviewer's concern regarding the novelty of our study. This work uniquely integrates computational modeling with experimental validation to elucidate the mechanistic relationship between *LIS1* mutations, ECM composition, and tissue mechanics. Unlike previous studies, our findings quantitatively demonstrate viscoelastic changes in human brain organoids across developmental stages and their correlation with ECM integrity. This approach provides a framework for understanding how *LIS1* mutations influence brain tissue mechanics.
- 2. ECM Alterations with MMP9 Treatment:** We recognize the reviewer's concern regarding the impact of MMP9 treatment on ECM structure. MMP9 is known to degrade key ECM components, including collagen, laminin, and fibronectin. While we do not claim permanent ECM remodeling, our RNA sequencing data reveal significant transcriptional changes in ECM-related genes within 10 minutes, demonstrating rapid cellular responses to the treatment. This supports the relevance of our findings in understanding ECM dynamics in *LIS1*-mutant organoids.
- 3. Brain Folding Effects:** We appreciate the opportunity to clarify this point. The brain organoid model used in this study is not well-suited for assessing cortical folding. However, in our previous work utilizing an on-chip brain organoid protocol, we demonstrated that *LIS1* mutations impact folding. Here, we focus specifically on how *LIS1* mutations influence ECM composition and tissue mechanics, showing that these alterations lead to significant changes in organoid stiffness and viscoelastic properties.

We believe that our clarifications address the reviewers' concerns and enhance the manuscript's clarity and scientific impact. We sincerely thank the reviewers and editors for their constructive input.